# Counterfactual Fairness by Combining Factual and Counterfactual Predictions

**Zeyu Zhou, Tianci Liu, Ruqi Bai, Jing Gao, Murat Kocaoglu, David I. Inouye**
Elmore Family School of Electrical and Computer Engineering
Purdue University
{zhou1059, liu3351, bai116, jinggao, mkocaoglu, dinouye}@purdue.edu

## Abstract

In high-stakes domains such as healthcare and hiring, the role of machine learning (ML) in decision-making raises significant fairness concerns. This work focuses on Counterfactual Fairness (CF), which posits that an ML model's outcome on any individual should remain unchanged if they had belonged to a different demographic group. Previous works have proposed methods that guarantee CF. Notwithstanding, their effects on the model's predictive performance remain largely unclear. To fill this gap, we provide a theoretical study on the inherent trade-off between CF and predictive performance in a model-agnostic manner. We first propose a simple but effective method to cast an optimal but potentially unfair predictor into a fair one with minimal performance degradation. By analyzing the excess risk incurred by perfect CF, we quantify this inherent trade-off. Further analysis on our method's performance with access to only incomplete causal knowledge is also conducted. Built upon this, we propose a practical algorithm that can be applied in such scenarios. Experiments on both synthetic and semi-synthetic datasets demonstrate the validity of our analysis and methods.

## 1 Introduction

Machine learning (ML) has been widely used in high-stakes domains such as healthcare [Daneshjou et al., 2021], hiring [Hoffman et al., 2018], criminal justice [Brennan et al., 2009], and loan assessment [Khandani et al., 2010], bringing with it critical ethical and social considerations. A prominent example is the bias observed in the COMPAS tool against African Americans in recidivism predictions [Brackey, 2019]. This issue is particularly alarming in an era where large-scale deep learning models, commonly trained on noisy data from the internet, are increasingly prevalent. Such models, due to their extensive reach and impact, amplify the potential for widespread and systemic biases. This increasing awareness underscores the need for ML practitioners to integrate fairness considerations into their work, extending their focus beyond merely maximizing prediction accuracy [Bolukbasi et al., 2016, Calders and Verwer, 2010, Dwork et al., 2012, Grgic-Hlaca et al., 2016, Hardt et al., 2016]. Various fairness notions have been developed, ranging from group-level measures such as group parity [Hardt et al., 2016] to individual-level metrics [Dwork et al., 2012]. Recently, there has been a growing interest in approaches based on causal inference, particularly in understanding the causal effects of sensitive attributes such as *gender* and *age* on decision-making [Chiappa, 2019, Galhotra et al., 2022, Khademi et al., 2019]. This has led to the proposal of Counterfactual Fairness (CF), which states that prediction for an individual in hypothetical scenarios where their sensitive attributes differ should remain unchanged [Kusner et al., 2017]. As an individual-level notion agnostic to the choice of similarity measure [Kusner et al., 2017, Rosenblatt and Witter, 2023], CF has recently

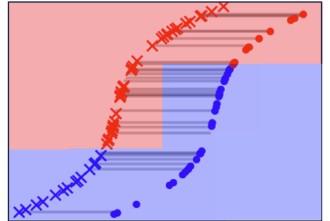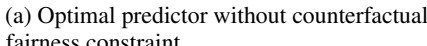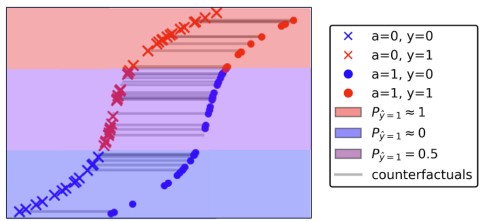

<table>
<tr><td>×</td><td>a=0, y=0</td></tr>
<tr><td>×</td><td>a=0, y=1</td></tr>
<tr><td>●</td><td>a=1, y=0</td></tr>
<tr><td>●</td><td>a=1, y=1</td></tr>
<tr><td></td><td>$P_{\hat{Y}=1} \approx 1$</td></tr>
<tr><td></td><td>$P_{\hat{Y}=1} \approx 0$</td></tr>
<tr><td></td><td>$P_{\hat{Y}=1} = 0.5$</td></tr>
<tr><td></td><td>counterfactuals</td></tr>
</table>

(a) Optimal predictor without counterfactual fairness constraint.

(b) Optimal predictor under perfect counterfactual fairness constraint.

Figure 1: The optimal (unfair) predictor (a) violates counterfactual fairness in the middle region because the predictions are different for the factual-counterfactual pairs[1] (denoted by line segments between $a = 0$ and $a = 1$). We prove that the optimal fair predictor (b) simply mixes the optimal unfair predictions at the factual and counterfactual points (i.e., mixes the predictions at both endpoints of the line). This mixing incurs the inherent excess risk associated with counterfactual fairness. Colors represent target classes ($Y$), and dot styles represent sensitive attributes ($A$).

gained traction [Anthis and Veitch, 2024, Nilforoshan et al., 2022, Makhlouf et al., 2022, Rosenblatt and Witter, 2023].

To achieve CF, Kusner et al. [2017] first proposed a naive solution, suggesting that predictions should only use non-descendants of the sensitive attribute in a causal graph. This approach only requires a causal topological ordering of variables and achieves perfect CF by construction. However, it limits the available features for downstream tasks and could be inapplicable in certain cases [Kusner et al., 2017]. For relaxation, they further proposed an algorithm that leverages latent variables. Extending this line of work, Zuo et al. [2023] introduced a technique that incorporates additional information by mixing factual and counterfactual samples. Although perfect CF has been established in their work, its predictiveness degraded, and whether the predictive power can be improved remains unknown. In parallel to this, another branch of research employed regularization and augmentation to *encourage* CF [Garg et al., 2019, Stefano et al., 2020, Kim et al., 2021]. However, as these methods cannot guarantee perfect CF, analyzing the optimal predictive performance under CF constraints is highly challenging.

To theoretically understand the tradeoff between CF and ML performance, we consider a class of invertible causal models and prove that the optimal solution under perfect Counterfactual Fairness (CF) has a simple form w.r.t. to the Bayes optimal classifier and explicitly quantify the excess risk of imposing a perfect CF constraint as has been done for non-causal fairness notions[Zhao and Gordon, 2022, Xian et al., 2023]. The optimal predictor under the fairness constraint can be achieved by combining factual and counterfactual predictions using a (potentially unfair) optimal predictor. Next, we quantify the excess risk between the optimal predictor with and without CF constraints. This quantity sheds light on the best possible model, in terms of predictive performance, under the stringent notion of perfect CF. Our results are illustrated in Figure 1. To consider scenarios with incomplete causal knowledge (e.g. unknown causal graph or model), we further study the CF and predictive performance degradation caused by imperfect counterfactual estimations. Inspired by our theoretical findings, we propose a plugin method that leverages a (potentially unfair) pretrained model to achieve a better tradeoff of fairness and predictive performance than the prior methods. Furthermore, we propose a method to improve the pretrained model that accounts for counterfactual estimation errors and can achieve good empirical performance even with limited causal knowledge. We summarize our contributions as follows:

1. We propose a CF method that is provably optimal in terms of predictive performance under perfect CF.

2. To the best of our knowledge, we are the first to characterize the inherent trade-off of CF and ML performance, which applies to all CF methods.

3. We investigate the CF and predictive performance degradation from estimation error resulting from limited causal knowledge and propose methods to mitigate estimation errors in practice.

4. We empirically demonstrate that our proposed CF methods outperform existing methods in both full and incomplete causal knowledge settings [2].

---

[1]Because of our invertibility assumption, any factual point has a unique counterfactual.

[2]Code can be found in https://github.com/inouye-lab/pcf

## 2 Preliminaries

**Notation** We use capital letters to represent random variables and lowercase letters to represent the realizations of random variables. Now we define a few variables that will be considered in this work. $A$ represents the sensitive attribute of an individual (e.g., gender), $Y$ represents the target variable to predict, $X$ represents observed features other than $A$ and $Y$, and $U$ represents unobserved confounding variables which are not caused by any observed variables while $a, y, x, u$ represent their realization respectively.

**Counterfactual** In this work, we use the framework of Structural Causal Models (SCMs) [Pearl, 2009]. A SCM is a triplet $\mathcal{M} = (\mathbf{U}, \mathbf{V}, \mathcal{F})$ where $\mathbf{U}$ represents exogenous variables (factors outside the model), $\mathbf{V}$ represents endogenous variables, and $\mathcal{F}$ contains a set of functions $F_i$ that map from $U_i$ and $Parent(V_i)$ to $V_i$. A counterfactual query asks a question like: what would the value of $Y$ be if $A$ had taken a different value given certain observations? For example, given that a person is a woman and given everything we observe about her performance in an interview, what is the probability of her getting the job if she had been a man? More formally, given a SCM, a counterfactual query can be written as $P(Y_{A=a}|W=w)$. Here $W = w$ is the evidence and $A = a$ in the subscript represents the intervention on $A$. For the general procedure to estimate counterfactuals, please refer to Pearl [2009].

**Counterfactual Fairness** Built upon the framework above, we focus on Counterfactual Fairness (CF), which requires the predictors to be fair among factual and counterfactual samples. More formally, it is defined as below

**Definition 2.1.** *(Counterfactual Fairness) We say a predictor $\hat{Y}$ is counterfactually fair if*

$$p(\hat{Y}_{A=a}|X = x, A = a) = p(\hat{Y}_{A=a'}|X = x, A = a), \quad \forall(x, a).$$

This definition states that intervention on $A$ should not affect the distribution of $\hat{Y}$. Using the same example above, the probability of a woman getting the job should be the same as that if she had been a man. For that goal, we use the following metric to evaluate CF

**Definition 2.2.** *(Total Effect) The Total Effect (TE) of a predictor $\hat{Y}$ is*

$$\text{TE} \triangleq \mathbb{E}[|\hat{Y}_{A=a} - \hat{Y}_{A=a'}|].$$

Therefore, a predictor is counterfactually fair if and only if TE = 0. Throughout the paper, we use TE to quantify the violation of counterfactual fairness.

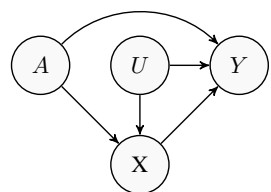

Figure 2: Causal graph. $A$ represents sensitive attribute, $Y$ represents the target variable, $U$ represents latent confounders, $X$ represents observed features. Note that the validity of our theoretical analysis holds for all causal models that satisfy the condition given by Assumption 3.1. It is not restricted to this specific graph.

## 3 Counterfactual Fairness via Output Combination

### 3.1 Problem Setup

We assume that all data we have is generated by a causal model [Pearl, 2009], and we consider the representative causal graph shown in Figure 2 that has been widely adopted in the fairness literature [Grari et al., 2023, Kusner et al., 2017, Zuo et al., 2023]. Our analysis is presented based on binary $A \in \{0, 1\}$ given its pivotal importance in the literature [Pessach and Shmueli, 2023] and for the sake of presentation clearness, but our analysis and our method can be naturally extended to multi-class $A$. We first state the main assumptions needed in this section.

**Assumption 3.1.**

1. *$A$ and $U$ are independent of each other.*

2. *The mapping between $X$ and $U$ is invertible given $A$.*

The first assumption is very common in the fairness literature. While the invertibility assumption might be restrictive in certain scenarios, it simplifies the theoretical analysis and has been adopted in

recent works on counterfactual estimation [Nasr-Esfahany et al., 2023, Zhou et al., 2024]. We expect that exact invertibility is not required in practice but rather only strong mutual information between $X$ and $U$ given $A$ would be sufficient. Further, we empirically validate the effectiveness of our method after relaxing invertibility in the experiment section. To facilitate our discussion, we first define $F_X$ as the mapping between $X$ and $(U, A)$, i.e., $X = F_X(U, A)$. According to our second assumption, $F_X(\cdot, a)$ is an invertible function, i.e., $\exists F_X^{*-1}, F_X^{*-1}(x, a) = F_X^{*-1}(F_X^*(u, a), a) = u, \forall (x, a)$. This assumption simplifies the counterfactual estimation of $X$ for different values of $A$ into a deterministic function. In our context, the counterfactual query is specifically $p(X_{a'}|X = x, A = a)$, which simplifies to a Dirac delta at a single value given the invertibility assumption. Thus, we introduce the concept of a deterministic counterfactual generating mechanism (CGM), denoted as $x_{a'} = G(x, a, a')$. Also, in our case, we will assume $A$ is binary so that $a$ and it's counterfactual $a'$ can be written as $1 - a$. All proofs can be found in Appendix A. Given this setup, the following lemma characterizes the perfect CF constraint on $\phi$.

**Lemma 3.2.** *Given Assumption 3.1, predictor $\phi$ on $(X, A)$ is counterfactually fair if and only if the predictor returns the same value for a sample and its counterfactuals, i.e.,* $\mathrm{TE}(\phi) = 0 \Leftrightarrow \phi(x, a) \stackrel{a.s.}{=} \phi(x_{1-a}, 1 - a), \quad \forall (x, a).$

The proof is straightforward from the definition of TE. Notably, this lemma helps disambiguate the question of whether counterfactual fairness is a distribution- or individual-level requirements as raised in Plecko and Bareinboim [2022]. In our setup, they are equivalent due to invertibility between $X$ and $U$ given $A$.

## 3.2 Optimal Counterfactual Fairness and Inherent Trade-off

Given the complete knowledge of the causal model, it is viable to satisfy the perfect CF constraint [Kusner et al., 2017, Zuo et al., 2023]. However, these methods are known to result in *empirical* degradation of ML models' performance, raising critical concerns about the *fairness-utility trade-offs*. Moreover, it is still unknown *to what extent the ML model performance has to be affected in order to achieve perfect CF*. In this section we provide a formal study on this to close the gap. Our solution consists of two steps. First, we propose a simple yet effective method that is provably optimal under the constraint of perfect CF. Next, we characterize the inherent trade-off between CF and predictive performance by checking the excess risk compared to a Bayes optimal (unfair) predictor. Our result shows that the inherent trade-off is dominated by the dependency between $Y$ and $A$, echoing previous analysis on non-causal based fairness notions [Chzhen et al., 2020, Xian et al., 2023]. For brevity, we refer to a *Bayes optimal* predictor as "optimal", and a model that satisfies *perfect CF* as "fair".

We start with the following theorem instantiating an *optimal and fair* predictor.

**Theorem 3.3.** *Given Assumption 3.1 [3] and loss $\ell$ (i.e., squared $L_2$ loss for regression tasks, and cross-entropy loss for classification tasks), an optimal and fair predictor (i.e., the best possible model(s) under the constraint of perfect CF) is given by the average of the optimal (potentially unfair) predictions on itself and all possible counterfactuals:*

$$\phi_{\mathrm{CF}}^*(x, a) \triangleq p(A = a)\phi^*(x, a) + p(A = 1-a)\phi^*(x_{1-a}, 1-a) \in \underset{\phi : \mathrm{TE}(\phi) = 0}{\operatorname{argmin}} \mathbb{E}[\ell(\phi(X, A), Y)],$$

*where $x_{1-a} = G^*(x, a, 1-a)$ is the counterfactual of $(x, a)$ when intervening with $A = 1-a$, and $\phi^*(x, a)$ is an unconstrained optimal predictor, i.e., $\phi^*(x, a) \triangleq \operatorname{argmin}_\phi \mathbb{E}[\ell(\phi(X, A), Y)] = \mathbb{E}[Y|X = x, A = a]$.*

This result suggests that, if we have to access to ground truth counterfactuals, a simple algorithm using a (potentially) unfair model could achieve strong fairness and accuracy. Built upon the above result, we are ready to characterize the inherent trade-off between CF and model performance by the following theorem.

**Theorem 3.4.** *The inherent trade-off between CF and predictive performance, characterized by the excess risk of the Bayes optimal predictor under the CF constraint, is given by*

$$\mathcal{R}_{CF}^* - \mathcal{R}^* = \sigma_A^2 \mathbb{E}_U \left[ \left( \mathbb{E}_{Y|U=u, A=a}[Y] - \mathbb{E}_{Y|U=u, A=1-a}[Y] \right)^2 \right],$$

---

[3]Note that while the discussion in this paper focuses on the causal model in Figure 2, our theorem is valid for more general cases that satisfy Assumption 3.1. And for brevity, the following discussion is done under this assumption unless otherwise stated.

*for regression tasks using squared $L_2$ loss where $\sigma_A^2$ denotes the variance of A; and*

$$\mathcal{R}_{CF}^* - \mathcal{R}^* = I(A; Y \mid U),$$

*for classification tasks using cross-entropy loss.*

Remarkably, the excess risks are completely characterized by the *inherent* dependency between $Y$ and $A$ as determined by the underlying causal mechanism, similar to non-causal based group fairness [Chzhen et al., 2020, Xian et al., 2023]. Moreover, they lower bound the excess risk of all possible predictors in order to achieve perfect CF.

### 3.3 Method with Incomplete Causal Knowledge

In this section, we aim to address CF in the scenario where causal knowledge is limited. Inspired by Theorem 3.3, we first present a simple plugin method as summarized in Algorithm 1. For regression tasks, $\hat{\mu}$ is the final output, and for classification tasks, $\hat{\mu}$ represents the probability of $Y = 1$, i.e., $p(Y = 1|X = x, A = a) = \mathbb{E}[Y|X = x, A = a]$. It is noteworthy that PCF is agnostic to the training of predictor $\phi$ that can be determined by the user freely. In fact, with access to the oracle CGM $G^*$, then PCF would achieve perfect CF as proved in the next result.

---

**Algorithm 1** Plug-in Counterfactual Fairness (PCF)

---

**Input:** Pretrained probabilistic prediction predictor $\phi : \mathcal{X} \times \mathcal{A} \to \mathcal{Y}$, CGM $G$, test datapoint $(x, a)$, prior distribution $p$ of $A$
**Output:** Predicted output $\hat{\mu}$
$\hat{x}_{1-a} \leftarrow G(x, a, 1 - a)$
$\hat{\mu} \leftarrow p(A = a)\phi(x, a) + p(A = 1 - a)\phi(\hat{x}_{1-a}, 1 - a)$

---

**Proposition 3.5.** *Given that $G$ is the ground truth counterfactual generating mechanism, i.e., $G(x, a, a') = x_{a'}, \forall(x, a, a')$, Algorithm 1 achieves perfect CF for any pretrained predictor $\phi$.*

Note that this proposition only requires access to ground truth $G^*$ and holds valid *for any pretrained predictor $\phi$*. If $\phi$ is further accurate, then the corresponding PCF is able to achieve high accuracy as well, which is empirically validated in the experiments.

#### 3.3.1 Given estimated $G$

Acquiring counterfactuals in practice can be a challenging task and could lead to estimation errors. In this section we provide a theoretical analysis on this. Specifically, the theorem below bounds the TE and excess risk due to the use of estimated counterfactuals.

**Theorem 3.6.** *Given an optimal predictor $\phi^*(x, a)$, suppose it is L-lipschitz continuous in $x$, and the counterfactual estimation error is bounded, i.e.,*

$$\max_{X,A} \|G^*(x_a, a, 1 - a) - \hat{G}(x_a, a, 1 - a)\|_2 \leq \varepsilon$$

*for some $\varepsilon \geq 0$, where $G^*$ and $\hat{G}$ represent the ground truth and estimated CGMs respectively. Then, the total effect (TE) of Algorithm 1 based on $\hat{G}$ is bounded by $L\varepsilon$. Moreover, for squared $L_2$ loss, the excess risk is bounded by $\sigma_A^2 L^2 \varepsilon^2 + 2\sigma_A^2 L\varepsilon \mathbb{E}_U[|\mathbb{E}[Y \mid U = u, A = 1] - \mathbb{E}[Y \mid U = u, A = 0]|]$, and for cross-entropy loss [4], the excess risk is bounded by $L\varepsilon$.*

Note that $\sigma_A$ and $\mathbb{E}_U[|\mathbb{E}[Y \mid U = u, A = 1] - \mathbb{E}[Y \mid U = u, A = 0]|]$ are inherent characteristic of the underlying mechanism and is independent of the counterfactual estimation. This suggests that if the counterfactuals are not too far away and $\phi^*$ is smooth, then fairness and prediction performance will not be significantly affected. In practice, CGM in Algorithm 1 can be obtained using counterfactual estimation methods, as discussed in Section 6.

---

[4] Here we assume the logits are L-lipschitz continuous in $x$.

### 3.3.2 Given estimated $G$ and $\phi$

In the previous section, we discussed how counterfactual estimation error directly impacts the performance of PCF in terms of CF and predictive performance. Here, we consider the situation where $\phi$ also needs to be estimated. We first note that the degradation in fairness remains the same as previous result

**Remark 3.7.** *The bound of TE given $\hat{\phi}$ and $\hat{G}$ follows that in Theorem 3.6.*

The proof is straightforward since the original proof in Theorem 3.6 does not use any characteristic of optimality. To achieve good predictive performance, a natural approach is to train $\phi$ on the observed data via Empirical Risk Minimization (ERM), which should fit the predictor well given sufficient samples and a reasonable predictor class. However, ERM can only approximate Bayes optimality within the support of the training data. Outside this support, its performance can deteriorate significantly, as extensively studied in areas such as Domain Adaptation [Farahani et al., 2021] and Domain Generalization [Zhou et al., 2022a]. Consequently, when integrated with an approximate $G$, we may encounter the issue where $p(\hat{X}_{A=1-a}|A = a) \neq p(X|A = 1 - a)$, inducing a distribution shift problem (Note that these would be equal given the graph in Figure 2 and Pearl's rules) [Kulinski and Inouye, 2023]. To mitigate this, we suggest improving $\phi$ on the estimated counterfactual distribution. More formally we define the following objective called Counterfactual Risk Minimization (CRM):

$$\min_{\phi} \mathbb{E}_{X,A,Y}[\ell(\phi(X,A),Y) + \ell(\phi(G(X,A,1-A),A),Y)]$$

This can be achieved either by augmenting the original training dataset or by fine-tuning with estimated counterfactual samples. The choice between training from scratch or fine-tuning depends on the scale of the experiment and computational constraints. It is important to note that the $Y$ corresponding to the estimated counterfactual should remain the same with that of the factual samples. While the optimal prediction for the counterfactual may differ from that of the original data, under the constraint of perfect CF, a predictor is required to predict the same outcome to counterfactual pairs. Hence, the optimal solution will change. This is exactly what causes the excess risk we characterized in Theorem 3.4. Furthermore, we can prove that, given the ground truth $G$, CRM yields the same optimal solution as PCF. Since this result is dependent on the ground truth $G$, we provide a more formal statement in Appendix B to ensure consistency.

In summary, this section discusses how to improve the estimation of $\phi$ under counterfactual estimation error. We propose using data augmentation or fine-tuning based on practical scenarios. Additionally, in domains with abundant off-the-shelf pre-trained models [Bommasani et al., 2021], we can potentially avoid this issue by using these models as a good proxy for $\phi$.

## 4 Related Works

**Fairness Notions** Fair Machine Learning has accumulated a vast literature that proposes various notions to measure fairness issues of machine learning models. Representative fairness notions can be categorized into three classes. *Group fairness*, such as demographic parity [Pedreshi et al., 2008] and equalized odds [Hardt et al., 2016], requires certain group-level statistical independence between model predictions and individuals' demographic information. Despite its conceptual simplicity, group fairness is known for ruling out perfect model performance [Hardt et al., 2016] and may allow for bias against certain individuals [Corbett-Davies et al., 2023]. *Individual fairness* [Dwork et al., 2012], on the other hand, asks a model to treat similar individuals similarly. However, determining the similarity between different individuals is often highly task-specific and open-ended. Recently, *counterfactual fairness* (CF, Kusner et al. [2017]) further takes the causal relationship of data attributes into consideration when measuring fairness. In words, counterfactual fairness proposes that a model should treat any individual the same as their *counterfactual* if the individual had been from another demographic group. As an individual-level notion agnostic to the choice of similarity measure [Kusner et al., 2017], CF has recently gained traction [Wu et al., 2019, Nilforoshan et al., 2022, Makhlouf et al., 2022, Rosenblatt and Witter, 2023]. Motivated by these recent advances, in this work, we focus on the counterfactual fairness.

**Methods for Fairness** Given an unfair dataset, attempts to achieve fairness fall into three categories. *Pre-processing* cleans the data before running machine learning models on it, typically by resampling

samples or removing undesired attributes [Kamiran and Calders, 2012]. *In-processing* intervenes the model-training process by incorporating fairness constraints [Zafar et al., 2017, Donini et al., 2018, Lohaus et al., 2020] or penalties [Mohler et al., 2018, Scutari et al., 2021, Liu et al., 2023]. *Post-processing* adjusts the raw model outputs to close the bias gap by, e.g., assigning each demographic group a unique decision threshold [Jang et al., 2022]. Post-processing has been favored as an efficient and practical solution because it does not require retraining the original model [Petersen et al., 2021, Xian et al., 2023]. To achieve CF, Kusner et al. [2017] applied pre-processing and discarded all descendants of the sensitive feature. Chen et al. [2024] pre-processed the data via orthogonalization and marginal distribution mapping. Garg et al. [2019], Stefano et al. [2020], Kim et al. [2021] in-processed the model training by penalizing CF violations but their solutions lack formal CF guarantees and often contain unsatisfactory bias after the intervention [Zuo et al., 2023]. Recently, Zuo et al. [2023] proposed another in-processing based solution that is capable of achieving perfect CF and better performance via mixing features. Ma et al. [2023] leveraged mediators estimated by Generative Adversarial Networks and provided a theoretical guarantee of CF under well-estimated counterfactuals. However, it is unclear whether their methods are optimal. Wang et al. [2023] leveraged predictor that satisfies equal counterfactual opportunity criterion to construct a counterfactually fair predictor. While they provide results on optimality, their findings assume an ideal setting where non-sensitive features are independent of sensitive features.

**Inherent Trade-off between Fairness and Predictiveness** Machine learning models are known to suffer from performance drops after fairness interventions [Hardt et al., 2016, Menon and Williamson, 2018, Chen et al., 2018], which is known as the *fairness-utility* trade-offs. Recently, inherent trade-offs towards non-causal based fairness such as demographic parity (DP) has been established separately for regression [Chzhen et al., 2020] and classification tasks [Xian et al., 2023]. The excess risks are characterized by certain distribution distance (i.e., Wasserstein-2 barycenter for regression, and total-variation or Wasserstein-1 barycenter for noiseless or noisy classification) between the conditional distribution of $Y$ given $A$. A similar trade-off between CF and predictiveness has also been empirically observed [Zuo et al., 2023]. Nonetheless, their inherent trade-off remains an open question. In this work we take the first step towards this goal and provide a quantitative analysis in both complete and incomplete causal knowledge settings as presented in Section 3. We hope our work sheds light on future works towards more effective CF.

# 5   Experiments

In this section, we validate our theorems and the effectiveness of our algorithms through experiments on synthetic and semi-synthetic datasets. On synthetic datasets, we focus on validating our theorems in settings where our assumptions hold. On semi-synthetic datasets, we aim to assess the effectiveness of our methods in more practical scenarios, where limited causal knowledge is available and the invertibility assumption is relaxed.

**Metrics**   We consider two metrics in this paper: Error and Total Effect (TE). The former evaluates whether each method can achieve its goal, irrespective of fairness. This is important because we can achieve perfect Counterfactual Fairness by always outputting fixed prediction given whatever input, but that is not useful at all. The latter is a common metric to evaluate Counterfactual Fairness [Kim et al., 2021, Zuo et al., 2023]. Given a test set $\mathcal{D}_{\text{test}}$, Error is defined as Error $= \frac{1}{|\mathcal{D}_{\text{test}}|} \sum_{x^{(i)} \in \mathcal{D}_{\text{test}}} \ell(\widehat{y}(x^{(i)}), y^{(i)})$ where $y^{(i)}$ is the ground truth target, $\widehat{y}(x^{(i)})$ is the prediction of $x^{(i)}$, and $\ell$ depends on the task. TE is defined as TE $= \frac{1}{|\mathcal{D}_{\text{test}}|} \sum_{x^{(i)} \in \mathcal{D}_{\text{test}}} |\widehat{y}(x^{(i)}) - \widehat{y}(x^{(i)}_{1-a})|$ where $x_{1-a}$ is the ground truth counterfactual corresponding to $x^{(i)}$. Since we only consider binary sensitive attribute, we further define TE$_0 = \frac{1}{|\{i:a^{(i)}=0\}|} \sum_{i:a^{(i)}=0} |\widehat{y}(x^{(i)}) - \widehat{y}(x^{(i)}_{1-a})|$ and TE$_1 = \frac{1}{|\{i:a^{(i)}=1\}|} \sum_{i:a^{(i)}=1} |\widehat{y}(x^{(i)}) - \widehat{y}(x^{(i)}_{1-a})|$ to evaluate Counterfactual Fairness for different group respectively.

**Methods**   In general, we consider the following methods: (1) **Empirical Risk Minimization (ERM):** Train a classifier on all features without any fairness consideration. Specifically $\widehat{y} = \phi(x, a)$, where $\phi$ represents the predictor. (2) **Counterfactual Fairness with** $U$ **(CFU) [Kusner et al., 2017]:** To achieve Counterfactual Fairness, CFU proposes to use $U$ for prediction. Specifically, $\widehat{y} = \phi(u)$. (3) **Counterfactual Fairness with fair representation (CFR) [Zuo et al., 2023]:** CFR

proposes to use $U$ and a symmetric version of $x, x_{1-a}$. Specifically, $\widehat{y} = \phi(\frac{x+x_{1-a}}{2}, u)$. (4) **Equal Counterfactual Opportunity (ECOCF)[Wang et al., 2023]:** An ECO predictor is adjusted to become counterfactually fair. Specifically, $\hat{y} = p(a)[p(a)\phi(x, a) + (1 - p(a))\phi(x, 1 - a)] + (1 - p(a))[(1 - p(a))\phi(x_{1-a}, 1 - a) + p(a)\phi(x_{1-a}, a)]$ (5) **PCF[5]:** As introduced in Algorithm 1, PCF mixes the output of factual and counterfactual prediction. Specifically, $\hat{y} = p(a)\phi(x, a) + (1 - p(a))\phi(x_{1-a}, 1 - a)$. (6) **PCF with analytic solution (PCF-Ana):** In synthetic experiments, instead of training via ERM, we can directly acquire bayes optimal $\phi$ in closed-form. Detailed can be found in Appendix C.2. (7) **PCF with CRM (PCF-CRM):** As discussed in Section 3.3, it could be hard to get the optimal predictor when there is counterfactual estimation error. Here due to the scale of our experiment, we augment the dataset with estimated counterfactuals rather than finetuning. Specifically, $\phi$ is trained via ERM on the dataset $\mathcal{D}_{\text{train}} = \{x^{(i)}, y^{(i)}, a^{(i)}\}_{i=1}^{N} \cup \{\hat{x}_{1-a}^{(i)}, y^{(i)}, 1 - a^{(i)}\}_{i=1}^{N}$.

## 5.1 Synthetic Dataset

In this section, we consider two regression synthetic datasets and two classification tasks where all of our assumptions in Assumption 3.1 are satisfied. The regression tasks are as below

*Linear-Reg*

$A \sim \text{Bernoulli}(p_A), U \sim \mathcal{N}(0, 1), \epsilon_Y \sim \mathcal{N}(0, 1)$

$X = w_A A + w_U U$

$Y = w_X X + w'_U U + w_Y \epsilon_Y$

*Cubic-Reg*

$A \sim \text{Bernoulli}(p_A), U \sim \mathcal{N}(0, 1), \epsilon_Y \sim \mathcal{N}(0, 1)$

$X = w_A A + w_U U$

$Y = w_X X^3 + w'_U U + w_Y \epsilon_Y$

The classification tasks take the same form except $Y \sim \text{Bernoulli}(\sigma(w_X X + w'_U U + w_Y \epsilon_Y)$ and $Y \sim \text{Bernoulli}(\sigma(w_X X^3 + w'_U U + w_Y \epsilon_Y))$ for *Linear-Cls* and *Cubic-Cls* respectively. More details could be found in Appendix C.1. Results are averaged over 5 different runs where the structural model is kept the same but data is resampled. All results shown in the main paper use KNN based predictor. Results with other predictors can be found in Appendix D.

**Optimality of PCF given true counterfactuals** We first test different methods in situations where all methods have access to ground truth counterfactuals and $U$ as needed. In Figure 3, we observe that while CFE, CFR and PCF all achieve perfect CF, PCF has lowest predictor error. This validates Theorem 3.3 regarding the optimality of PCF under the constraint of CF. Furthermore, since here ERM can get solution close to optimal predictor (this indicates the plugin $\phi$ used by PCF is also close to being optimal), we can also observe the inherent fairness-utility trade-off discussed in Theorem 3.4.

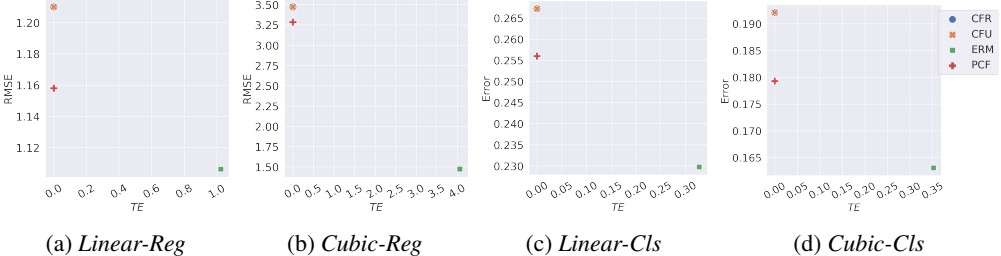

(a) *Linear-Reg*     (b) *Cubic-Reg*     (c) *Linear-Cls*     (d) *Cubic-Cls*

Figure 3: Results on synthetic datasets given ground truth counterfactuals.

**Performance under controllable error** Here we investigate a more practical scenario where both counterfactuals and $U$ need to be estimated. To investigate how error and TE changes with counterfactual estimation error in a more controllable way and investigate , we simulate the estimation error by adding gaussian noise. Specifically, $\hat{x}_{a'} = x_{a'} + \epsilon$ and $\hat{u} = u + \epsilon$ where $\epsilon \sim \mathcal{N}(\beta, \alpha)$. In Figure 4, we observe that while the fairness and ML performance (especially fairness) of CFE, CFR and PCF tends to get worse as error gets more significant, PCF remains best for all noise level.

**Investigating source of error** Here we further investigate what could be source of error in the previous scenario. As discussed in Section 3.3.2, in practice, two things in Theorem 3.3 break down:

---

[5]Essentially PCF with ERM (PCF-ERM). For brevity, we just call it PCF.

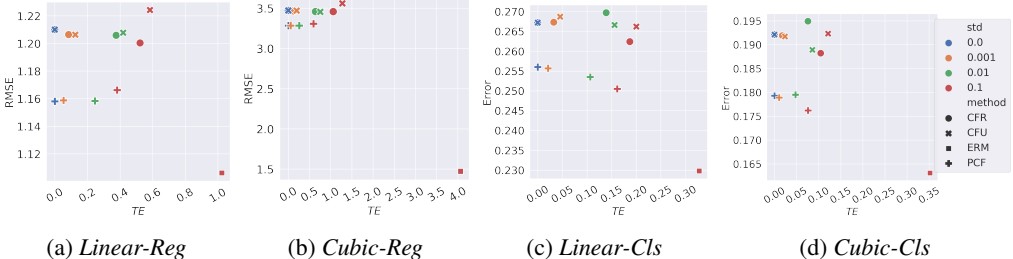

| (a) *Linear-Reg* | (b) *Cubic-Reg* | (c) *Linear-Cls* | (d) *Cubic-Cls* |

Figure 4: Results on synthetic datasets under counterfactual estimation error. Different color represents different $\alpha$ indicating the standard deviation of the error ($\epsilon \sim \mathcal{N}(0, \alpha)$) while shape represents different algorithms. Results with different $\beta$ can be found in Appendix D.

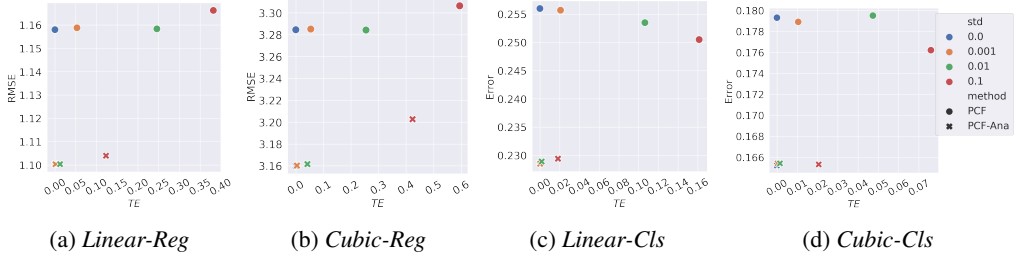

| (a) *Linear-Reg* | (b) *Cubic-Reg* | (c) *Linear-Cls* | (d) *Cubic-Cls* |

Figure 5: Results on synthetic datasets comparing PCF and PCF-Analytic. Different color represents different $\alpha$ indicating the standard deviation of the error ($\epsilon \sim \mathcal{N}(0, \alpha)$) while shape represents different algorithms. Results with different $\beta$ can be found in Appendix D.

access to Bayes optimal classifier and ground truth counterfactuals. In Figure 5, we observe that PCF-Analytic tends to be more robust against counterfactual estimation error than PCF. We argue this is because $\phi$ used in PCF is not trained well on the estimated counterfactual distribution.

## 5.2 Semi-synthetic Dataset

In this section, we consider Law School Success dataset [Wightman, 1998] where the sensitive attribute is gender and the target is first-year grade.. The main goal of this experiment is to validate the effectiveness of our methods in more practical scenarios where limited causal knowledge is available and the invertibility assumption is relaxed.

To compute TE, we need access to ground truth counterfactuals. Hence we train a generative model on real dataset to generate semi-synthetic dataset following the method in Zuo et al. [2023]. We want to emphasize that counterfactuals are hidden from downstream models and used for the evaluation of TE only. This way, we get access to the ground truth $u^*$ and can generate ground

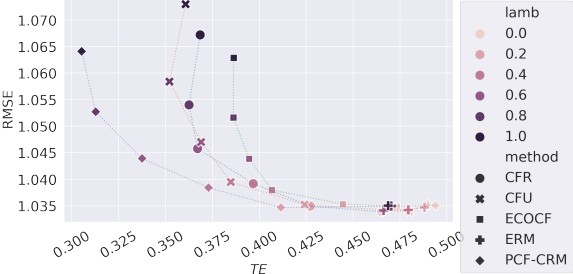

Figure 6: Results on Sim-Law with estimated counterfatuals. The predictor is a MLP regressor. We also test the convex combination of each algorithm and ERM. For example, PCF-CRM with $\lambda$ means $\hat{y} = \lambda\hat{y}_{\text{PCF-CRM}} + (1 - \lambda)\hat{y}_{\text{ERM}}$. This suggests that PCF-CRM can achieve lower Error given the same TE and lower TE given the same Error.

truth counterfactuals without any error. In our investigation, exogenous noise, factual data and counterfactual data are all actually the simulated version of original datasets. However they do follow a fixed data generating mechanism that is close to the real data. More details could be found in Appendix C.1. All experiments are repeated 5 times on the same semi-synthetic dataset.

**Results** In Figure 6, we observe that PCF-CRM achieves better CF and lower Error in comparison to CFU and CFR. This validates our improvement Section 3.3.2 indeed leads to more practical algorithm. Results on comparing PCF and PCF-CRM can be found in Appendix D, which further justifies this. While ERM could achieve lower error, it has worst fairness. This is inevitably determined by the inherent trade-off discussed in Theorem 3.4. Furthermore, inspired by the trade-off, we test the result of mixing all predictors with ERM. The curve shows that PCF-CRM remains optimal given fixed CF and best CF given fixed error. This demonstrates again that PCF-CRM is the best among all methods.

# 6 Conclusion and Discussion

**Conclusion** In this work, we conducted a formal investigation of the trade-off between Counterfactual Fairness (CF) and predictive performance. We proved that combining factual and counterfactual predictions with a potentially unfair, optimal predictor achieves optimal CF. Additionally, we derived the excess risk between predictors with and without CF constraints, quantifying the minimum performance degradation necessary to ensure perfect CF. To address incomplete causal knowledge, we analyzed the effects of imperfect counterfactual estimations on CF and predictive performance. We proposed a plugin approach that leverages pre-trained models for optimal fair prediction and developed a practical method to mitigate estimation errors.

Despite the theoretical contributions of our method, two limitations may impact practical applicability: (1) access to ground-truth counterfactuals and (2) access to Bayes optimal predictors. Below, we delve into these limitations, clarifying how our methods can be practically applied and how they can benefit from contributions from the broader community. We hope this discussion will also inspire future research directions.

**Access to ground truth counterfactuals** While how to better estimate counterfactuals is out of the scope of this work, it is indeed an unavoidable challenge faced by the community of Counterfactual Fairness. It not only limits the deployment of CF algorithms, but also leads to difficulty in validating proposed CF methods. While counterfactual data can be obtained in specific scenarios, such as through randomized controlled trials, it is challenging to acquire in most applications. There are some works in the field of causality that aims at estimating counterfactuals. For instance, Nasr-Esfahany et al. [2023] proves counterfactual identifiability under certain causal graphs. However, in more general scenarios, such causal knowledge may be lacking and identifying the causal graph itself can be challenging. These tasks have been well studied in the field of causal discovery [Chickering, 2002, Colombo et al., 2014] and causal representation learning [Schölkopf et al., 2021]. Solutions to this problem typically rely on strong assumptions, such as the linearity of Structural Causal Models (SCMs) or additive noise [Shimizu et al., 2006, Hoyer et al., 2008, Peters et al., 2014]. More recently, Zhou et al. [2024] propose a method of estimating counterfactuals without the need to identify the causal model or graph. We believe this approach to direct counterfactual estimation could have the potential to be a good plugin counterfactual estimator in our algorithm. Additionally, generative models could also be used to generate samples as if they had come from a different sensitive attribute [Choi et al., 2018, Zhou et al., 2022b, 2023, Rombach et al., 2022]. These methods often offer the advantage of higher sample quality, especially in modalities such as images or natural language. However, they must be applied with considerable care, as they generally lack integration with the causal model and may introduce significant estimation errors.

**Access to Bayes optimal predictors** Another crucial plugin estimator of our method is the optimal predictor. In classical ML settings, achieving a good estimator for the counterfactual distribution often requires retraining or fine-tuning. However, in this era, with the abundance of pre-trained models, such as foundation models [Bommasani et al., 2021], it could be much easier to get a predictor that is close to being optimal. Rather, given that these models are trained on noisy internet data and have extensive reach and impact, it is of great importance to find effective ways to debias them. We propose that our plugin algorithm could be a suitable solution due to its post-processing nature, which avoids incurring significant computational costs.

## Acknowledgement

Z.Z., R.B., and D.I. acknowledge support from NSF (IIS-2212097), ARL (W911NF-2020221), and ONR (N00014-23-C-1016). M.K. acknowledges support from NSF CAREER 2239375, IIS 2348717, Amazon Research Award and Adobe Research. T.L. and J.G. acknowledge support from NSF-IIS2226108. Any opinions, findings, and conclusions or recommendations expressed in this material are those of the author(s) and do not necessarily reflect the views of any funding source.

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

# A Proofs

## A.1 Proof of Lemma 3.2

*Proof of Lemma 3.2.*

$$\text{TE}(\phi) = 0 \Leftrightarrow \mathbb{E}[|\phi(X, A) - \phi(X_{1-A}, 1 - A)|] = 0 \Leftrightarrow \phi(x, a) \overset{\text{a.s.}}{=} \phi(x_{1-a}, 1 - a), \quad \forall(x, a), \tag{1}$$

where the first equality is by definition and the second equality is because absolute value is always non-negative for any $(x, a)$. Thus, the predictions must be almost surely equal for all $(x, a)$. Similarly, if they are all equal on the non-zero metric set, then the expectation must be 0. □

## A.2 Proof of Theorem 3.3

Before proving the main theorem, we first provide one well-known lemma that reminds the reader of the well-known result of the optimal predictor, which is denoted by $\phi^*$ in the theorem statement.

**Lemma A.1** (Optimal Predictor is Conditional Mean). *The conditional mean $\mathbb{E}[Y|X = x]$ is the optimal predictor without fairness constraints for classification with cross-entropy loss and for regression with MSE loss.*

*Proof.* First, let's establish that the optimal predictor without constraints is in fact $\mathbb{E}[Y|X = x]$. For squared $L_2$ loss, we have that derivative:

$$\mathbb{E}[\ell(\phi(X), Y)]$$
$$= \mathbb{E}_X[\mathbb{E}_{Y|X}[(Y - \phi(X))^2]]$$
$$= \mathbb{E}_X[\mathbb{E}_{Y|X}[Y^2] - 2\mathbb{E}_{Y|X}[Y\phi(X)] + \mathbb{E}_{Y|X}[\phi(X)^2]]$$
$$= \mathbb{E}_X[\mathbb{E}_{Y|X}[Y^2] - 2\phi(X)\mathbb{E}_{Y|X}[Y] + \phi(X)^2\mathbb{E}_{Y|X}[1]]$$
$$= \mathbb{E}_X[\mathbb{E}_Y[Y^2] - 2\phi(X)\mathbb{E}[Y|X] + \phi(X)^2]$$

Taking the derivative of the inside expectation w.r.t. $\phi(X)$ and setting to 0 yields $\phi^*(X) = \mathbb{E}[Y|X]$.

Now let's look at cross-entropy loss for classification:

$$\mathbb{E}[\ell(\phi(X), Y)]$$
$$= \mathbb{E}_X[\mathbb{E}_{Y|X}[-Y\log(\phi(X)) - (1 - Y)\log(1 - \phi(X))]]$$
$$= \mathbb{E}_X[-\log(\phi(X))\mathbb{E}[Y|X] - \log(1 - \phi(X))\mathbb{E}[(1 - Y)|X]]$$

Again, if you take the derivative w.r.t. $\phi(X)$ and set to 0, we see that $\phi^*(X) = \mathbb{E}[Y|X]$. □

Now we seek to prove Theorem 3.3.

*Proof.* First, we decompose the factual error across the sensitive attribute $A$ given the exogenous noise $U$.

$$\mathbb{E}_{X,A,Y}[\ell(\phi(X, A), Y)]$$
$$= \mathbb{E}_{U,A,Y}[\ell(\phi(F_X^*(U, A), A), Y)]$$
$$= \mathbb{E}_U[\mathbb{E}_A[\mathbb{E}_{Y|U,A}[\ell(\phi(F_X^*(U, A), A), Y)]]]$$
$$= \mathbb{E}_U[p(A = a)\mathbb{E}_{Y|U,A=a}[\ell(\phi(F_X^*(U, a), a), Y)] + p(A = 1 - a)\mathbb{E}_{Y|U,A=1-a}[\ell(\phi(F_X^*(U, 1 - a), 1 - a), Y)]].$$

Consider $U = u$, inside the expectation we have

$$p(A = a)\mathbb{E}_{Y|U=u,A=a}[\ell(\phi(F_X^*(u, a), a), Y)] + p(A = 1 - a)\mathbb{E}_{Y|U=u,A=1-a}[\ell(\phi(F_X^*(u, 1 - a), 1 - a), Y)]$$
$$= p(A = a)\mathbb{E}_{Y|X=x,A=a}[\ell(\phi(x, a), Y)] + p(A = 1 - a)\mathbb{E}_{Y|X=x_{1-a},A=1-a}[\ell(\phi(x_{1-a}, 1 - a), Y)],$$

where w.l.o.g., $x$ is viewed as the factual and $x_{a'}$ is viewed as the counterfactual. Because of invertibility, these two terms are unique for every $(u, a)$ or correspondingly $(x, a)$ combination and thus the problem decomposes across $U$. Thus, the factual loss can be viewed as a combination of the factual loss from one specific $a$ plus the counterfactual loss for $a'$ for each point $x$.

We have the following subproblems indexed by $u$: The factual loss can be viewed as a combination of the factual loss from one specific $a$ plus the counterfactual loss for $1 - a$ for each point $x$. Notice that the constraint is $\phi(x, a) \overset{a.s.}{=} \phi(x_{1-a}, 1 - a)$ from Lemma 3.2. We can directly push the constraint into the optimization problem by optimizing over $\phi_0 \triangleq \phi(x, a) \overset{a.s.}{=} \phi(x_{1-a}, 1 - a)$:

$$\underset{\phi}{\operatorname{argmin}} \, p(A = a)\mathbb{E}_{Y|X=x,A=a}[\ell(\phi_0, Y)] + p(A = 1 - a)\mathbb{E}_{Y|X=x_{1-a},A=1-a}[\ell(\phi_0, Y)] \quad (2)$$

Taking $\ell$ as squared $L_2$ loss: we have

$$\underset{\phi}{\operatorname{argmin}} \, p(A = a)\mathbb{E}_{Y|X=x,A=a}[(Y - \phi_0)^2] + p(A = 1 - a)\mathbb{E}_{Y|X=x_{1-a},A=1-a}[(Y - \phi_0)^2]$$

$$= \underset{\phi}{\operatorname{argmin}} \, p(A = a)\{\mathbb{E}_{Y|X=x,A=a}[Y^2] - 2\phi_0\mathbb{E}_{Y|X=x,A=a}[Y] + \phi_0^2\}$$

$$+ p(A = 1 - a)\{\mathbb{E}_{Y|X=x_{1-a},A=1-a}[Y^2] - 2\phi_0\mathbb{E}_{Y|X=x_{1-a},A=1-a}[Y] + \phi_0^2\}.$$

Similarly, if we take $\ell$ as (binary) cross-entropy loss: we have

$$\underset{\phi}{\operatorname{argmin}} \, p(A = a)\mathbb{E}_{Y|X=x,A=a}[-(Y \log(\phi) + (1 - Y) \log(1 - \phi))]$$

$$+ p(A = 1 - a)\mathbb{E}_{Y|X=x_{1-a},A=1-a}[-(Y \log(\phi) + (1 - Y) \log(1 - \phi))]$$

It is simple to see that both loss functions are convex, thus could obtain a unique solution by taking the derivative. Thus, for each $x, x_{a'}$ induced by $U = u$, we could get the optimal $\phi_0$:

$$\phi_0 = \sum_{a' \in \{0,1\}} p(A = a')\phi^*(x_{a'}, a'),$$

where $\phi^*$ is the optimal predictor from the lemma above. This result holds for every $u$ and thus gives the final result. $\square$

## A.3 Proof of Theorem 3.4

*Proof.* Let $\phi^*(x, a)$ and $\phi_{\text{CF}}^*(x, a)$ be the Bayes optimal predictor under no constraint and CF constraint respectively. We have shown that $\phi^*(x, a) = \mathbb{E}_{Y|X=x,A=a}[Y]$ and $\phi_{\text{CF}}^*(x, a) = p(A = a)\phi^*(x, a) + p(A = 1 - a)\phi^*(x_{1-a}, 1 - a)$.

Noting that $\phi^*$ is Bayes optimal, its risk satisfies $\mathcal{R}^* \leq \mathcal{R}_{\text{CF}}^*$ where $\mathcal{R}_{\text{CF}}^*$ denotes the risk of $\phi_{\text{CF}}^*(x, a)$. By definition, the excess risk of $\phi_{\text{CF}}^*(x, a)$ is

$$\mathcal{R}_{\text{CF}}^* - \mathcal{R}^* = \mathbb{E}_{X,A}\left[\mathbb{E}_{Y|X=x,A=a}[\ell(\phi_{\text{CF}}^*(x, a), Y) - \ell(\phi^*(x, a), Y)]\right].$$

For regression task and real-valued $Y$, we take $\ell$ as squared $L_2$ loss and have

$$\mathcal{R}_{\text{CF}}^* - \mathcal{R}^*$$

$$= \mathbb{E}_{X,A}\left[\mathbb{E}_{Y|X=x,A=a}[(\phi_{\text{CF}}^*(x, a) - Y)^2 - (\phi^*(x, a) - Y)^2]\right]$$

$$= \mathbb{E}_{X,A}\left[\mathbb{E}_{Y|X=x,A=a}[(\phi_{\text{CF}}^*(x, a) - \phi^*(x, a))(\phi_{\text{CF}}^*(x, a) + \phi^*(x, a) - 2Y)]\right]$$

$$= \mathbb{E}_{X,A}\left[(\phi_{\text{CF}}^*(x, a) - \phi^*(x, a))\mathbb{E}_{Y|X=x,A=a}[(\phi_{\text{CF}}^*(x, a) + \phi^*(x, a) - 2Y)]\right]$$

$$= \mathbb{E}_{X,A}\left[(\phi_{\text{CF}}^*(x, a) - \phi^*(x, a))^2\right]$$

$$= \mathbb{E}_{X,A}\left[((p(A = a)\phi^*(x, a) + p(A = 1 - a)\phi^*(x_{1-a}, 1 - a) - \phi^*(x, a))^2\right]$$

$$= \mathbb{E}_{X,A}\left[(1 - p(A = a))^2(\phi^*(x, a) - \phi^*(x_{1-a}, 1 - a))^2\right]$$

$$= \mathbb{E}_{X,A}\left[p^2(A = 1 - a)(\phi^*(x, a) - \phi^*(x_{1-a}, 1 - a))^2\right]$$

$$= \mathbb{E}_A\left[p^2(A = 1 - a)\mathbb{E}_{X|A=a}[(\phi^*(x, a) - \phi^*(x_{1-a}, 1 - a))^2]\right].$$

Let's define

$$\Delta_a \triangleq \mathbb{E}_{X|A=a}\left[\left(\mathbb{E}_{Y|X=x,A=a}[Y] - \mathbb{E}_{Y|X=x_{1-a},A=1-a}[Y]\right)^2\right]$$

$$\overset{(a)}{=} \mathbb{E}_{U|A=a}\left[\left(\mathbb{E}_{Y|U=u,A=a}[Y] - \mathbb{E}_{Y|U=u,A=1-a}[Y]\right)^2\right]$$

$$\overset{(b)}{=} \mathbb{E}_U\left[\left(\mathbb{E}_{Y|U=u,A=a}[Y] - \mathbb{E}_{Y|U=u,A=1-a}[Y]\right)^2\right] = \Delta_{1-a},$$

where $(a)$ holds from the invertibility between $X$ and $U$ given $A$, and $(b)$ holds from the fact that $U$ and $A$ are independent. For simplicity, we denote $\Delta = \Delta_a = \Delta_{1-a}$. Notably, $\Delta$ measures the expected change of $Y$ due to the change of $A$ over all possible $U$, and is in fact a measure of their dependency. Furthermore, we have

$$
\begin{aligned}
&\mathbb{E}_A\left[p^2(A = 1 - a)\mathbb{E}_{X|A=a}[(\phi^*(x, a) - \phi^*(x_{1-a}, 1 - a))^2]\right] \\
=&\mathbb{E}_A\left[p^2(A = 1 - a)\Delta_a\right] \\
=&\sum_a p(A = a)p^2(A = 1 - a)\Delta_a \\
=&p(A = 0)p^2(A = 1)\Delta_0 + p(A = 1)p^2(A = 0)\Delta_1 \\
=&p(A = 0)p(A = 1)\Delta \\
=&\sigma_A^2\Delta.
\end{aligned}
$$

Next, for classification task and binary $Y$ using cross-entropy loss, then

$$
\begin{aligned}
&\mathcal{R}_{\text{CF}}^* - \mathcal{R}^* \\
=&\mathbb{E}_{X,A}\left[\mathbb{E}_{Y|X=x,A=a}\left[-Y\log\phi_{\text{CF}}^*(x, a) - (1 - Y)\log(1 - \phi_{\text{CF}}^*(x, a))\right] + \right. \\
&\left. \mathbb{E}_{Y|X=x,A=a}\left[Y\log\phi^*(x, a) + (1 - Y)\log(1 - \phi^*(x, a))\right]\right] \\
=&\mathbb{E}_{X,A}\left[-\phi^*\log\phi_{\text{CF}}^* - (1 - \phi^*)\log(1 - \phi_{\text{CF}}^*) + \phi^*\log\phi^* + (1 - \phi^*)\log(1 - \phi^*)\right] \\
=&\mathbb{E}_{X,A}\left[\phi^*\log\frac{\phi^*}{\phi_{\text{CF}}^*} + (1 - \phi^*)\log\frac{1 - \phi^*}{1 - \phi_{\text{CF}}^*}\right] \\
\overset{(a)}{=}&\mathbb{E}_{X,A}\left[D_{KL}[p(Y \mid X, A)\|\mathbb{E}_A[p(Y \mid X_A, A)]]\right] \\
\overset{(b)}{=}&\mathbb{E}_{U,A}\left[D_{KL}[p(Y \mid U, A)\|\mathbb{E}_A[p(Y \mid U, A)]]\right] \\
=&\mathbb{E}_{U,A}\left[D_{KL}[p(Y \mid U, A)\|p(Y \mid U)]\right] \\
=&\mathbb{E}_U\mathbb{E}_A\left[D_{KL}[p(Y \mid U, A)\|p(Y \mid U)]\right] \\
=&I(A; Y \mid U),
\end{aligned}
$$

where $(a)$ holds from noting that $\phi^*(x, a) = p(Y = 1 \mid X = x, A = a)$ and $\phi_{CF}^* = \mathbb{E}_A[p(Y = 1 \mid X = x_A, A)]$, and $(b)$ again holds from the invertibility between $X$ and $U$ given $A$. $\qquad\square$

### A.4 Proof of Proposition 3.5

*Proof.*

$$
\begin{aligned}
\mathbb{E}[\hat{Y}|X = x, A = a] &= \hat{\mu}(x, a) \\
&= p(A = a)\phi(x_a, a) + p(A = 1 - a)\phi(G(x, a, 1 - a), 1 - a) \\
&= p(A = a)\phi(G(x_{1-a}, 1 - a, a), a) + p(A = 1 - a)\phi(x_{1-a}, 1 - a) \\
&= \hat{\mu}(x_{1-a}, 1 - a) \\
&= \mathbb{E}[\hat{Y}|X = x_{1-a}, A = 1 - a],
\end{aligned}
$$

where the middle qualities are by the properties of the invertible and ground truth CGM. Because the factual output for the algorithm is the same as the counterfactual output, then the TE must be 0 by Lemma 3.2. $\qquad\square$

### A.5 Proof of Theorem 3.6

*Proof.* We first bound TE.

Let $x_{a \to 1-a} \triangleq G^*(x_a, a, 1-a)$, we have

$$
\begin{aligned}
\text{TE} &= \mathbb{E}_{X,A}\left[|\phi_{\text{PCF}}(x_a, a) - \phi_{\text{PCF}}(x_{a \to 1-a}, 1-a)|\right] \\
&= \mathbb{E}_{X,A}\big[|p(A=a)\phi(x_a, a) + p(A=1-a)\phi(\hat{x}_{a \to 1-a}, 1-a) \\
&\qquad -p(A=a)\phi(\hat{x}_{1-a \to a}, a) - p(A=1-a)\phi(x_{a \to 1-a}, 1-a)|\big] \\
&= \mathbb{E}_{X,A}\Big[|p(A=a)\phi(x_a, a) + p(A=1-a)\phi(\hat{G}(x_a, a, 1-a), 1-a) \\
&\qquad -p(A=a)\phi(\hat{G}(G^*(x_a, a, 1-a), 1-a, a) - p(A=1-a)\phi(G^*(x_a, a, 1-a), 1-a)|\Big] \\
&\overset{(a)}{\leq} \mathbb{E}_{X,A}\Big[p(A=a)|\phi(x_a, a) - \phi(\hat{G}(G^*(x_a, a, 1-a), 1-a, a), a)| \\
&\qquad +p(A=1-a)|\phi(\hat{G}(x_a, a, 1-a), 1-a) - \phi(G^*(x_a, a, 1-a), 1-a)|\Big] \\
&= \mathbb{E}_{X,A}\Big[p(A=a)|\phi(G^*(G^*(x_a, a, 1-a), 1-a, a), a) - \phi(\hat{G}(G^*(x_a, a, 1-a), 1-a, a), a)| \\
&\qquad +p(A=1-a)|\phi(\hat{G}(x_a, a, 1-a), 1-a) - \phi(G^*(x_a, a, 1-a), 1-a)|\Big] \\
&\overset{(b)}{\leq} \mathbb{E}_{X,A}\Big[p(A=a)L|G^*(G^*(x_a, a, 1-a), 1-a, a) - \hat{G}(G^*(x_a, a, 1-a), 1-a, a)| \\
&\qquad +p(A=1-a)L|\hat{G}(x_a, a, 1-a) - G^*(x_a, a, 1-a)|\Big] \\
&\overset{(c)}{\leq} \mathbb{E}_{X,A}\left[p(A=a)L\varepsilon + p(A=1-a)L\varepsilon\right] \\
&= L\varepsilon.
\end{aligned}
$$

Here $(a)$ holds by the convexity of absolute value, $(b)$ is from the L-lipschitz property of $\phi$, and $(c)$ is by the bound of counterfactual estimation error.

Now we prove the bound for the error. Taking $\ell$ as squared $L_2$ loss, we have

$$
\begin{aligned}
\mathcal{R} &= \mathbb{E}_{X,A}\mathbb{E}_{Y|X=x, A=a}\left[(\phi_{\text{PCF}}(x, a) - y)^2\right] \\
&= \mathbb{E}_{X,A}\mathbb{E}_{Y|X=x, A=a}\left[(\phi_{\text{PCF}}(x, a) - \phi^*_{\text{CF}}(x, a) + \phi^*_{\text{CF}}(x, a) - y)^2\right],
\end{aligned}
$$

Taking the inner expectation and omit subscript for brevity, we have

$$
\begin{aligned}
&\mathbb{E}\left[(\phi_{\text{PCF}}(x, a) - \phi^*_{\text{CF}}(x, a) + \phi^*_{\text{CF}}(x, a) - y)^2\right] \\
&= \mathbb{E}\left[(\phi_{\text{PCF}}(x, a) - \phi^*_{\text{CF}}(x, a))^2\right] + 2\mathbb{E}\left[(\phi_{\text{PCF}}(x, a) - \phi^*_{\text{CF}}(x, a))(\phi^*_{\text{CF}}(x, a) - y)\right] + C \\
&= p(A=1-a)^2(\phi^*(\hat{x}_{1-a}, 1-a) - \phi^*(x_{1-a}, 1-a))^2 \\
&\quad + 2p(A=1-a)(\phi^*(\hat{x}_{1-a}, 1-a) - \phi^*(x_{1-a}, 1-a))(\phi^*_{\text{CF}}(x, a) - \phi^*(x, a)) + C \\
&\leq p(A=1-a)^2(\phi^*(\hat{x}_{1-a}, 1-a) - \phi^*(x_{1-a}, 1-a))^2 \\
&\quad + 2p(A=1-a)|\phi^*(\hat{x}_{1-a}, 1-a) - \phi^*(x_{1-a}, 1-a)(\phi^*_{\text{CF}}(x, a) - \phi^*(x, a)| + C \\
&\overset{(a)}{\leq} p(A=1-a)^2 L^2\varepsilon^2 + 2p(A=1-a)L\varepsilon|\phi^*_{\text{CF}}(x, a) - \phi^*(x, a)| + C,
\end{aligned}
$$

where $C$ denotes the remaining term that only depends on $\phi^*_{\text{CF}}$. Here $(a)$ holds from the fact that the counterfactual estimation error is bounded by $\varepsilon$ and the assumption that $\phi^*$ is $L$-lipschitz.

Next, take the outer expectation,

$$
\begin{aligned}
\mathcal{R} &\leq \mathbb{E}_A[p(A=1-a)^2]L^2\varepsilon^2 + 2L\varepsilon\mathbb{E}_{X,A}[p(A=1-a)|\phi^*_{\text{CF}}(x, a) - \phi^*(x, a)|] + \mathcal{R}^*_{\text{CF}} \\
&= \sigma_A^2 L^2\varepsilon^2 + 2L\varepsilon\mathbb{E}_{X,A}[p(A=1-a)^2|\phi^*(x_{1-a}, 1-a) - \phi^*(x, a)|] + \mathcal{R}^*_{\text{CF}}.
\end{aligned}
$$

Note that taking expectation of $C$ with respect to the joint distribution of $X, A$ is in fact the optimal risk $\mathcal{R}_{\text{CF}}^*$. Reorganization gives us

$$
\begin{aligned}
&\mathcal{R} - \mathcal{R}_{\text{CF}}^* \\
\leq{}& \sigma_A^2 L^2 \varepsilon^2 + 2L\varepsilon \mathbb{E}_{X,A}[p(A=1-a)^2|\phi^*(x_{1-a}, 1-a) - \phi^*(x,a)|] \\
={}& \sigma_A^2 L^2 \varepsilon^2 + 2L\varepsilon \mathbb{E}_{X,A}[p(A=1-a)^2|\mathbb{E}[Y \mid X = x_{1-a}, A = 1-a] - \mathbb{E}[Y \mid X = x, A = a]|] \\
={}& \sigma_A^2 L^2 \varepsilon^2 + 2L\varepsilon \mathbb{E}_{U,A}[p(A=1-a)^2|\mathbb{E}[Y \mid U = u, A = 1-a] - \mathbb{E}[Y \mid U = u, A = a]|] \\
={}& \sigma_A^2 L^2 \varepsilon^2 + 2L\varepsilon \mathbb{E}_U[p(A=1)p(A=0)^2|\mathbb{E}[Y \mid U = u, A = 0] - \mathbb{E}[Y \mid U = u, A = 1]| \\
&+ p(A=0)p(A=1)^2|\mathbb{E}[Y \mid U = u, A = 1] - \mathbb{E}[Y \mid U = u, A = 0]|] \\
={}& \sigma_A^2 L^2 \varepsilon^2 + 2\sigma_A^2 L\varepsilon \mathbb{E}_U[|\mathbb{E}[Y \mid U = u, A = 1] - \mathbb{E}[Y \mid U = u, A = 0]|].
\end{aligned}
$$

When $\ell$ is cross-entropy loss, the excess risk is

$$
\begin{aligned}
&\mathcal{R} - \mathcal{R}_{\text{CF}}^* \\
={}& \mathbb{E}_{X,A} \mathbb{E}_{Y|X=x,A=a}\left[-Y \log \frac{\phi_{\text{PCF}}}{\phi_{\text{CF}}^*} - (1-Y)\log \frac{1 - \phi_{\text{PCF}}}{1 - \phi_{\text{CF}}^*}\right]
\end{aligned}
$$

Here we assume the logit (i.e., the inverse function of sigmoid) is $L$-lipschitz continuous in $x$, i.e.,

$$
|f^*(x, a) - f^*(\hat{x}, a)| \leq L\|x - \hat{x}\|, \forall x, \hat{x}, a
$$

where $\phi^* \triangleq \sigma \circ f^*$ and $f^*(x, a) = \log \frac{\phi^*(x,a)}{1-\phi^*(x,a)}$. Now we check the excess risk. The first term can be upper bounded by

$$
\begin{aligned}
Y \log \frac{\phi_{\text{CF}}^*}{\phi_{\text{PCF}}} &= Y \log \frac{p(A=a)\phi^*(x,a) + p(A=1-a)\phi^*(x_{1-a}, 1-a)}{p(A=a)\phi^*(x,a) + p(A=1-a)\phi^*(\hat{x}_{1-a}, 1-a)} \\
&\leq Y(\phi_{\text{CF}}^*)^{-1}\left(p(A=a)\phi^*(x,a)\log \frac{p(A=a)\phi^*(x,a)}{p(A=a)\phi^*(x,a)}\right. \\
&\quad\left. + p(A=1-a)\phi^*(x_{1-a}, 1-a)\log \frac{p(A=1-a)\phi^*(x_{1-a}, 1-a)}{p(A=1-a)\phi^*(\hat{x}_{1-a}, 1-a)}\right) \\
&= Y \frac{p(A=1-a)\phi^*(x_{1-a}, 1-a)}{\phi_{\text{CF}}^*} \log \frac{\phi^*(x_{1-a}, 1-a)}{\phi^*(\hat{x}_{1-a}, 1-a)} \\
&= Y C_1 \log \frac{\phi^*(x_{1-a}, 1-a)}{\phi^*(\hat{x}_{1-a}, 1-a)},
\end{aligned}
$$

where the inequality holds by applying log sum inequality. For the inequality, it is derived from

$$
\begin{aligned}
\phi_{\text{CF}}^* \log \frac{\phi_{\text{CF}}^*}{\phi_{\text{PCF}}} &\leq p(A=a)\phi^*(x,a)\log \frac{p(A=a)\phi^*(x,a)}{p(A=a)\phi^*(x,a)} \\
&+ p(A=1-a)\phi^*(x_{1-a}, 1-a)\log \frac{p(A=1-a)\phi^*(x_{1-a}, 1-a)}{p(A=1-a)\phi^*(\hat{x}_{1-a}, 1-a)}
\end{aligned}
$$

And $C_1$ is defined as below

$$
C_1 = \frac{p(A=1-a)\phi^*(x_{1-a}, 1-a)}{P(A=a)\phi^*(x,a) + P(A=1-a)\phi^*(x_{1-a}, 1-a)}
$$

Similarly,

$$
\begin{aligned}
(1-Y)\log \frac{1 - \phi_{\text{CF}}^*}{1 - \phi_{\text{PCF}}} &= (1-Y)\log \frac{p(A=a) - p(A=a)\phi^*(x,a) + p(A=1-a) - p(A=1-a)\phi^*(x_{1-a}, 1-a)}{p(A=a) - p(A=a)\phi^*(x,a) + p(A=1-a) - p(A=1-a)\phi^*(\hat{x}_{1-a}, 1-a)} \\
&\leq (1-Y)\frac{p(A=1-a)(1 - \phi^*(x_{1-a}, 1-a))}{1 - \phi_{\text{CF}}^*} \log \frac{1 - \phi^*(x_{1-a}, 1-a)}{1 - \phi^*(\hat{x}_{1-a}, 1-a)} \\
&= (1-Y)C_2 \log \frac{1 - \phi^*(x_{1-a}, 1-a)}{1 - \phi^*(\hat{x}_{1-a}, 1-a)}.
\end{aligned}
$$

where

$$C_2 = \frac{p(A = 1 - a)(1 - \phi^*(x_{1-a}, 1 - a))}{1 - \phi^*_{\text{CF}}}$$

$$= \frac{p(A = 1 - a)(1 - \phi^*(x_{1-a}, 1 - a))}{1 - P(A = a)\phi^*(x, a) - P(A = 1 - a)\phi^*(x_{1-a}, 1 - a)}$$

$$= \frac{p(A = 1 - a)(1 - \phi^*(x_{1-a}, 1 - a))}{p(A = a) - P(A = a)\phi^*(x, a) + p(A = 1 - a)(1 - \phi^*(x_{1-a}, 1 - a))}$$

Put together

$$Y \log \frac{\phi^*_{\text{CF}}}{\phi_{\text{PCF}}} + (1 - Y) \log \frac{1 - \phi^*_{\text{CF}}}{1 - \phi_{\text{PCF}}}$$

$$\leq Y C_1 \log \frac{\phi^*(x_{1-a}, 1 - a)}{\phi^*(\hat{x}_{1-a}, 1 - a)} + (1 - Y)C_2 \log \frac{1 - \phi^*(x_{1-a}, 1 - a)}{1 - \phi^*(\hat{x}_{1-a}, 1 - a)}$$

$$\leq \left| Y C_1 \log \frac{\phi^*(x_{1-a}, 1 - a)}{\phi^*(\hat{x}_{1-a}, 1 - a)} + (1 - Y)C_2 \log \frac{1 - \phi^*(x_{1-a}, 1 - a)}{1 - \phi^*(\hat{x}_{1-a}, 1 - a)} \right|$$

$$\overset{(a)}{\leq} \left| Y C_1 \log \frac{\phi^*(x_{1-a}, 1 - a)}{\phi^*(\hat{x}_{1-a}, 1 - a)} - (1 - Y)C_2 \log \frac{1 - \phi^*(x_{1-a}, 1 - a)}{1 - \phi^*(\hat{x}_{1-a}, 1 - a)} \right|$$

$$\overset{(b)}{\leq} \max \{ Y C_1, (1 - Y)C_2 \} \left| \log \frac{\phi^*(x_{1-a}, 1 - a)}{\phi^*(\hat{x}_{1-a}, 1 - a)} - \log \frac{1 - \phi^*(x_{1-a}, 1 - a)}{1 - \phi^*(\hat{x}_{1-a}, 1 - a)} \right|$$

$$\leq \left| \log \frac{\phi^*(x_{1-a}, 1 - a)}{\phi^*(\hat{x}_{1-a}, 1 - a)} - \log \frac{1 - \phi^*(x_{1-a}, 1 - a)}{1 - \phi^*(\hat{x}_{1-a}, 1 - a)} \right|$$

$$= \left| \log \frac{\phi^*(x_{1-a}, 1 - a)}{1 - \phi^*(x_{1-a}, 1 - a)} - \log \frac{\phi^*(\hat{x}_{1-a}, 1 - a)}{1 - \phi^*(\hat{x}_{1-a}, 1 - a)} \right| \leq L \| x_{1-a} - \hat{x}_{1-a} \|.$$

Step $(a)$ holds from the observation that the two $\log$ terms must have different signs: unless $\phi^*(x_{1-a}, 1 - a) = \phi^*(\hat{x}_{1-a}, 1 - a) = 0.5$, otherwise it is impossible to have both

$$\phi^*(x_{1-a}, 1 - a) \geq \phi^*(\hat{x}_{1-a}, 1 - a)$$
$$1 - \phi^*(x_{1-a}, 1 - a) \geq 1 - \phi^*(\hat{x}_{1-a}, 1 - a),$$

hold simultaneously. Step $(b)$ holds from the fact that the two terms now have the same sign so we can safely upper bound them, and this maximum is upper bounded by 1. Finally, the excess risk

$$\mathbb{E}_{X,A} \mathbb{E}_{Y|X=x,A=a} \left[ Y \log \frac{\phi^*_{cf}}{\phi_{pcf}} + (1 - Y) \log \frac{1 - \phi^*_{cf}}{1 - \phi_{pcf}} \right] \leq \mathbb{E}_{X,A} \mathbb{E}_{Y|X=x,A=a} \left[ L \| x_{1-a} - \hat{x}_{1-a} \| \right] \leq L\varepsilon,$$

so long as $\| x_{1-a} - \hat{x}_{1-a} \| \leq \varepsilon$. This completes the proof.

# B    Counterfactual Risk Minimization

**Theorem B.1.** *Given that $G$ is the ground truth counterfactual generating mechanism, CRM and PCF will have the same optimal solution under the constraint of perfect Counterfactual Fairness, i.e.,*

$$\underset{\phi:\text{TE}(\phi)=0}{\text{argmin}} \ \mathbb{E}_{X,A,Y}[\ell(\phi(X, A), Y)] = \underset{\phi:\text{TE}(\phi)=0}{\text{argmin}} \ \mathbb{E}_{X,A,Y}[\ell(\phi(X, A), Y) + \ell(\phi(G(X, A, 1 - A), A), Y)]$$

*Proof.*

$$\mathbb{E}_{X,A,Y}[\ell(\phi(X,A),Y) + \ell(\phi(G(X,A,1-A),1-A),Y)]$$

$$=\mathbb{E}_{U,A,Y}[\ell(\phi(F_X^*(U,A),A),Y) + \ell(\phi(G(F_X^*(U,A),A,1-A),1-A),Y)]$$

$$=\mathbb{E}_{U,A,Y}[\ell(\phi(F_X^*(U,A),A),Y) + \ell(\phi(F_X^*(U,1-A),1-A),Y)]$$

$$=\mathbb{E}_U[p(A=a)\mathbb{E}_{Y|U=u,A=a}[\ell(\phi(F_X^*(u,a),a),Y)] + p(A=1-a)\mathbb{E}_{Y|U=u,A=1-a}[\ell(\phi(F_X^*(u,1-a),1-a),Y)]$$

$$+ p(A=a)\mathbb{E}_{Y|U=u,A=a}[\ell(\phi(F_X^*(u,1-a),1-a),Y)] + p(A=1-a)\mathbb{E}_{Y|U=u,A=1-a}[\ell(\phi(F_X^*(u,a),a),Y)]]$$

$$=\mathbb{E}_X[p(A=a)\mathbb{E}_{Y|X=x,A=a}[\ell(\phi(x,a),Y)] + p(A=1-a)\mathbb{E}_{Y|X=x_{1-a},A=1-a}[\ell(\phi(x_{1-a},1-a),Y)]$$

$$+ p(A=a)\mathbb{E}_{Y|X=x,A=a}[\ell(\phi(x_{1-a},1-a),Y)] + p(A=1-a)\mathbb{E}_{Y|X=x_{1-a},A=1-a}[\ell(\phi(x,a),Y)]]$$

Enforcing the constraint of CF, we define $\phi_0 \triangleq \phi(x,a) = \phi(x_{1-a},1-a)$, then the optimization problem of the inner expectation becomes

$$\underset{\phi_0}{\operatorname{argmin}} p(A=a)\mathbb{E}_{Y|X=x,A=a}[\ell(\phi_0,Y)] + p(A=1-a)\mathbb{E}_{Y|X=x_{1-a},A=1-a}[\ell(\phi_0,Y)]$$

$$+ p(A=a)\mathbb{E}_{Y|X=x,A=a}[\ell(\phi_0,Y)] + p(A=1-a)\mathbb{E}_{Y|X=x_{1-a},A=1-a}[\ell(\phi_0,Y)]$$

$$= \underset{\phi_0}{\operatorname{argmin}} 2p(A=a)\mathbb{E}_{Y|X=x,A=a}[\ell(\phi_0,Y)] + 2p(A=1-a)\mathbb{E}_{Y|X=x_{1-a},A=1-a}[\ell(\phi_0,Y)]$$

where the objective is just a scaled version of that in the proof of Theorem 3.3. Hence, we get the same minimizer. □

## C   Experiment Details

We included the codes to reproduce our results. All GPU related experiments are run on RTX A5000.

### C.1   Dataset

**Synthetic Dataset**   In this section, we consider the two regression synthetic datasets and two classification tasks where all of our assumptions in Assumption 3.1 are satisfied.

*Linear-Reg*

$A \sim \text{Bernoulli}(p_A), U \sim \mathcal{N}(0,1), \epsilon_Y \sim \mathcal{N}(0,1)$

$X = w_A A + w_U U$

$Y = w_X X + w_U' U + w_Y \epsilon_Y$

*Cubic-Reg*

$A \sim \text{Bernoulli}(p_A), U \sim \mathcal{N}(0,1), \epsilon_Y \sim \mathcal{N}(0,1)$

$X = w_A A + w_U U$

$Y = w_X X^3 + w_U' U + w_Y \epsilon_Y$

where in our experiments the parameters are chosen as $w_A = 1, w_X = 1, w_Y = 1, w_U = 1, w_U' = 1$.

We also consider the following two classification tasks

*Linear-Cls*

$A \sim \text{Bernoulli}(p_A), U \sim \mathcal{N}(0,1), \epsilon_Y \sim \mathcal{N}(0,1)$

$X = w_A A + w_U U$

$Y \sim \text{Bernoulli}(\sigma(w_X X + w_U' U + w_Y \epsilon_Y))$

*Cubic-Cls*

$A \sim \text{Bernoulli}(p_A), U \sim \mathcal{N}(0,1), \epsilon_Y \sim \mathcal{N}(0,1)$

$X = w_A A + w_U U$

$Y \sim \text{Bernoulli}(\sigma(w_X X^3 + w_U' U + w_Y \epsilon_Y))$

where in our experiments the parameters are chosen as $w_A = 2, w_X = 1, w_Y = 1, w_U = 1, w_U' = 1$.

**Semi-synthetic Dataset**   We consider Law School Success [Wightman, 1998]. The sensitive attribute is gender and the target is first-year grade. Other features contain race, LSAT and GPA. However, since we need to evaluate TE of each method which requires access to ground truth, we use the simulated version of those datasets. Following a similar setup in [Zuo et al., 2023], we train a generative model to get semi-synthetic datasets. Specifically, we train a VAE with the following structure $u \sim Enc(x,a), x \sim Dec_1(u,a), y \sim Dec_2(u,x)$. The training objective includes a normal VAE objective to reconstruct $x$ via $Enc$ and $Dec_1$, and a supervised objective to generate $y$ via $Dec_2$. After training, we first sample a prior $u \sim \mathcal{N}(0,I)$ and $a \sim Bernoulli(p)$ (where $p$ is acquired based on empirical frequency in real data), then we get the semi-synthetic $x, y$ using $Dec_1$ and $Dec_2$. We want to emphasize that counterfactuals, regardless of train or test set, are hidden from downstream models and used for evaluation only. This way, we get access to the ground truth $u^*$ and can generate ground truth counterfactuals without any error. In our investigation, exogenous noise, factual data and counterfactual data are all actually the simulated version of original datasets. However they do follow a fixed data generating mechanism that is close to the real data.

## C.2 Analytic Solution on Synthetic Datasets

We know the analytic solution of Bayes optimal predictor in our synthetic data experiments. More specifically, for *Linear-Reg*, we have

$$\phi^*(x,a) = \mathbb{E}[Y|X=x, A=a] = w_X x + \frac{w'_U}{w_U}(x - w_A a)$$

For *Cubic-Reg*, we have

$$\phi^*(x,a) = \mathbb{E}[Y|X=x, A=a] = w_X x^3 + \frac{w'_U}{w_U}(x - w_A a)$$

For *Linear-Cls*, we have

$$\phi^*(x,a) = \mathbb{E}[Y|X=x, A=a] = \sigma(w_X x + \frac{w'_U}{w_U}(x - w_A a))$$

For *Cubic-Cls*, we have

$$\phi^*(x,a) = \mathbb{E}[Y|X=x, A=a] = \sigma(w_X x^3 + \frac{w'_U}{w_U}(x - w_A a))$$

## C.3 Prediction Models

In our synthetic experiments, we mainly use KNN based predictors. We use the default parameters in `scikit-learn`. All MLP methods uses a structure with hidden layer $(20, 20)$ and Tanh activation.

In semi-synthetic experiments, we use MLP methods uses a structure with hidden layer $(5, 5)$ and Tanh activation as this is closer to the ground truth SCM.

# D Additional Results

## D.1 Additional results on synthetic datasets

In Figure 7, we test how how all algorithms perform when using ground truth counterfactuals and $U$ on additional type of predictors. We observe that PCF achieves lower error than CFU and CFR, which is similar to what we observe in Figure 3. This further validates our theory regarding optimality of PCF.

In Figure 8, following the investigation in Figure 4, we test with adding gaussian noise with different mean. We observe that when it is a fixed bias, CFU and CFR achieves better fairness than PCF. Though PCF still achieves best predictive performance. Furthermore, as we increase variance of the noise, PCF outperform these two methods in terms of both fairness and ML performance. In Figure 9, similar to Figure 5, we observe PCF-Analytic significantly improves over PCF. Notably, it is not affected by bias as PCF.

## D.2 Additional results on semi-synthetic datasets

In Figure 10, we included the expanded version of Figure 6 with $\text{TE}_0$ and $\text{TE}_1$. We observe that they show a very similar trend. In Figure 11, we directly compare PCF (with ERM) and PCF-CRM. The results validate the necessity of CRM as a plugin estimator $\phi$ in the case of limited causal knowledge.

## D.3 Additional experiments

To evaluate the performance of our method across a broader range of data generating mechanisms, counterfactual estimation models, and datasets, we conducted experiments using the Disentangled Causal Effect Variational Autoencoder (DCEVAE) on the Adult dataset [Asuncion et al., 2007]. Similarly, we trained one DCEVAE to simulate the Sim-Adult dataset and another DCEVAE to estimate counterfactuals. Here we followed the data preprocessing, model setup, and hyperparameter choices specified in Zuo et al. [2023]. A key difference between the setup used here and our setup on the Law School dataset is that, in this case, the encoder of the ground truth DCEVAE takes $y$ as an input. This modification introduces an inconsistency between the ground truth CGM and the estimated CGM. In Figure 12, we observe a trend similar to that in Figure 6, which indicates the effectiveness of PCF-CRM.

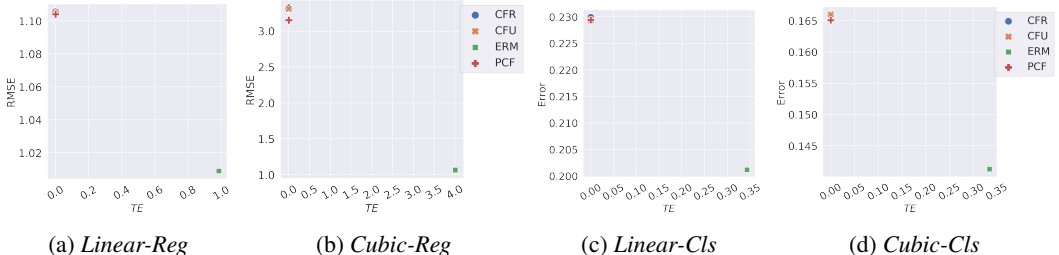

(a) *Linear-Reg*  (b) *Cubic-Reg*  (c) *Linear-Cls*  (d) *Cubic-Cls*

Figure 7: Results on synthetic datasets given ground truth counterfactuals with MLP predictors.

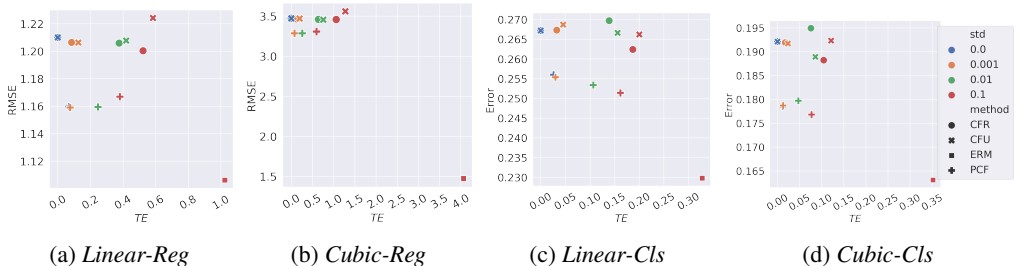

(a) *Linear-Reg*  (b) *Cubic-Reg*  (c) *Linear-Cls*  (d) *Cubic-Cls*

Figure 8: Results on synthetic datasets under counterfactual estimation error with KNN predictors. Different color represents different $\alpha$ indicating the standard deviation of the error ($\epsilon \sim \mathcal{N}(0.001, \alpha)$) while shape represents different algorithms.

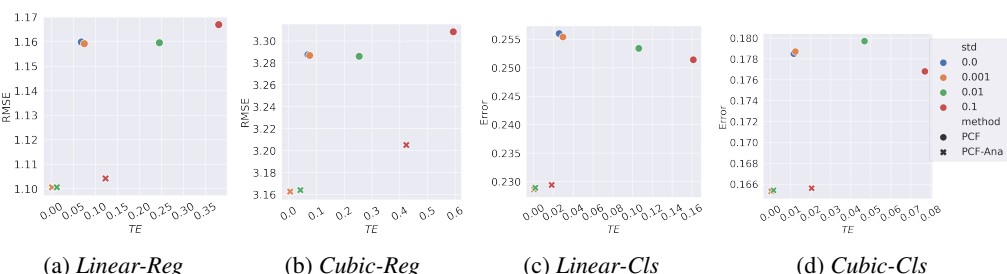

(a) *Linear-Reg*  (b) *Cubic-Reg*  (c) *Linear-Cls*  (d) *Cubic-Cls*

Figure 9: Results on synthetic datasets comparing PCF and PCF-Analytic with KNN predictors. Different color represents different $\alpha$ indicating the standard deviation of the error ($\epsilon \sim \mathcal{N}(0.001, \alpha)$) while shape represents different algorithms.

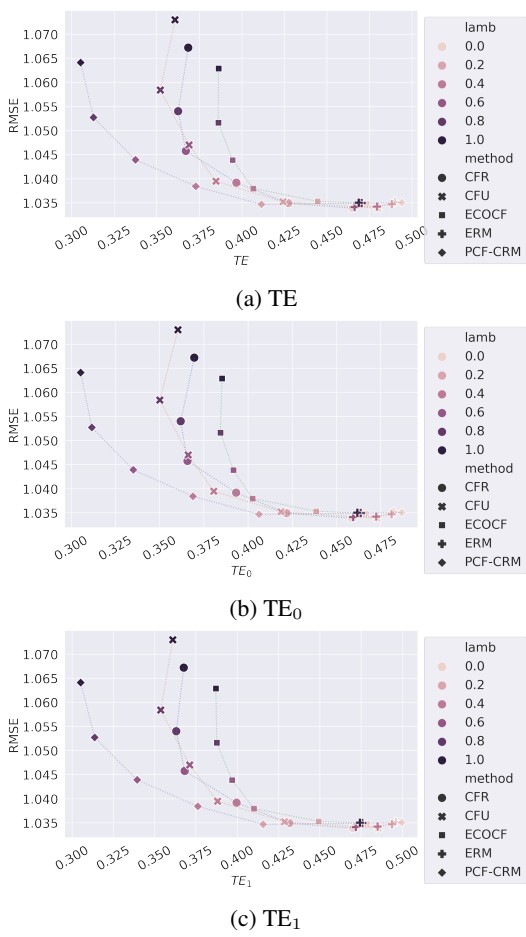

(a) TE

(b) TE$_0$

(c) TE$_1$

Figure 10: Results on Sim-Law with estimated counterfatuals. The predictor is a MLP regressor. We also test the convex combination of each algorithm and ERM. For example, PCFAug with $\lambda$ means $\hat{y} = \lambda\hat{y}_{\text{PCFAug}} + (1 - \lambda)\hat{y}_{\text{ERM}}$. This suggests that PCFAug can achieve lower Error given the same TE and lower TE given the same Error.

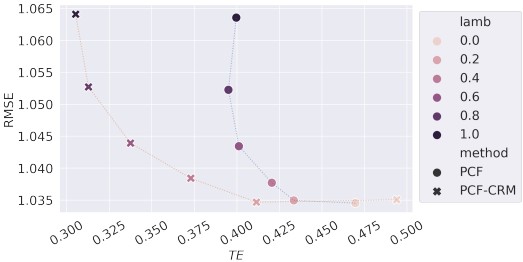

Figure 11: Comparison between PCF and PCF-CRM on Sim-Law with estimated counterfatuals. The predictor is a MLP regressor.

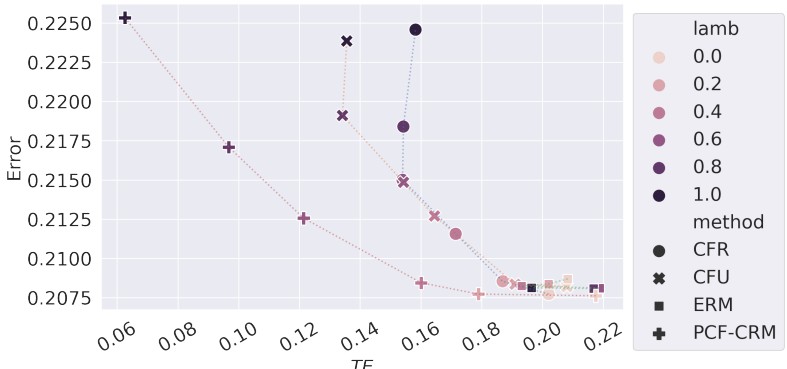

Figure 12: Results on Sim-Adult with estimated counterfatuals. The predictor is a MLP classifier.

