# OpenReview forum: "Counterfactual Fairness by Combining Factual and Counterfactual Predictions"
_NeurIPS.cc/2024/Conference — NeurIPS 2024 poster_

### Official Review · Reviewer_e7ax · 2024-07-11

**Soundness:** 3
**Presentation:** 3
**Contribution:** 3
**Rating:** 7
**Confidence:** 4

**Summary:**

This work studies the problem of counterfactual fairness in ML predictions. The authors first prove the form of the best possible fair predictor in a model-agnostic manner. Then, the excess risk of optimal predictor under the CF constraint is characterized. Based on the theoretical findings, the authors propose a post-preprocessing algorithm to achieve CF given arbitrary ML predictor. The authors perform comprehensive numerical studies to demonstrate the performance of the proposed algorithm compared to several baselines.

**Strengths:**

1. The theoretical analyses throughout the paper are technically sound and comprehensive.

2. Presentation of the work is easy to follow and the writing quality is high.

**Weaknesses:**

1. Although the authors acknowledges that the access to ground truth counterfactuals is out of the scope of this work, I personally think it is better to incorporate existing counterfactual estimation procedures into the Algorithm 1, which would enhance its practical applicability for the researchers and policy makers.

2. The authors can include more existing methods in the numerical experiments for comparison, such as the post-processing method by Wang et. al [1] and the pre-processing method by Chen et. al [2], to see the performance gap between these algorithms.


[1] Wang, Yixin, Dhanya Sridhar, and David Blei. "Adjusting Machine Learning Decisions for Equal Opportunity and Counterfactual Fairness." Transactions on Machine Learning Research (2023).

[2] Chen, Haoyu, et al. "On learning and testing of counterfactual fairness through data preprocessing." Journal of the American Statistical Association (2023): 1-11.

**Questions:**

1. See Weakness 2.

2. Wang et. al [1] proposed a weighted average CF predictor and discussed the optimality of the proposed predictor, which shares similar spirit with this work. The authors can discuss the difference between this work and . Wang et. al [1]'s work in terms of accuracy and fairness guarantee.

3. Line 183; I am confused about the intuition about the objective function for counterfactual risk minimization (CRM). In my opinion, the counterfactual outcome may not be necessarily close to factual outcome $Y$. Can the authors explain more about the reason why adding the second term in the objective function?


[1] Wang, Yixin, Dhanya Sridhar, and David Blei. "Adjusting Machine Learning Decisions for Equal Opportunity and Counterfactual Fairness." Transactions on Machine Learning Research (2023).

**Limitations:**

The authors discuss the limitation of restrictive irreversible assumption in section 3 and access to ground truth counterfactuals and optimal predictors in section 6.

---

> ### Author Rebuttal · Authors · 2024-08-06
>
> We thank the reviewer for acknowledging the writing and theoretical contribution of our paper. We will address each of your concerns and questions below.
>
> >**Incorporate existing counterfactual estimation procedures into the Algorithm 1**
>
> We will follow your suggestion and modify the Algorithm 1 to an end-to-end solution by incorporating the counterfactual estimation procedure. Our experiments did use estimated counterfactuals. as detailed in Section 5.2. In particular, we trained a VAE to estimate counterfactuals to validate our method’s practical applicability.
>
> >**Comparison with [1]**
>
> We thank the reviewer for bringing this interesting work to our attention. After carefully reading through the paper, we note the following key differences compared to our work.
>
> 1. The two *algorithms are different*.  For the sake of simple illustration, consider the case with binary sensitive attribute $A$ and deterministic counterfactuals (i.e., $G(x, a, a')$ is a deterministic function). The algorithm in [1], according to eq (2) therein, is given by
> $$f_{ecocf}(x,a) = p(a)f_{eco}(x) + p(1-a)f_{eco}(x_{1-a}).$$
> In words, [1] constructs the fair predictor by taking the (weighted) average over equal counterfactual opportunity (ECO)-fair predictions $f_{eco}$. In contrast, our algorithm,
> $$f_{pcf}= p(a)f(x,a) + p(1-a)f(x_{1-a},1-a),$$
> directly averages the original ML predictors $f$ that may be potentially unfair.
>
> 2. The *optimality conditions* are different:
> In essences, in [1] optimality holds only in an ideal world where non-sensitive features $X$ and sensitive features $A$ are independent of each other. Our work, on the other hand, is not restricted to such special case. Mathematically speaking, the expected loss in [1] is over $P(A)P(X)$, while ours is over $P(A,X)$.
>
> 3. The *problem setups* are different: the discussions in [1] are mainly done without consideration of hidden confounders, while our discussion explicitly considers the existence of an unobserved confounder $U$.
>
> Besides the above conceptual difference, we also found notable empirical performance gaps between the two method, which is shown in Fig2 of the above rebuttal pdf, and is detailed in the following response.
>
> >**Experiment with additional baselines**
>
> Following the reviewer’s suggestion, we conduct additional experiments and compare the proposed method with [1] and [2]. The results are shown in Figure 2 in the attached pdf in Response to All. We note that *our method outperforms these two baselines*. Due to time and space constraints, more comprehensive study will be added in the revised version.
>
>
> >**Discussion of Counterfactual Risk Minimization**
>
> We wholeheartedly agree that the optimal prediction for the counterfactual might be different from that of the original data. However, under the constraint of Counterfactual Fairness (CF), a predictor needs to predict the same outcome for counterfactual pairs. Hence, the optimal solution will change. This is exactly what causes the excess risk we characterized in Theorem 3.4. Moreover, we could actually prove that the theoretic minimum of CRM optimization is actually the optimal CF solution. We will provide a more formal analysis in the final manuscript.
>
> [1] Wang, Yixin, Dhanya Sridhar, and David Blei. "Adjusting Machine Learning Decisions for Equal Opportunity and Counterfactual Fairness." Transactions on Machine Learning Research (2023).
>
> [2] Chen, Haoyu, et al. "On learning and testing of counterfactual fairness through data preprocessing." Journal of the American Statistical Association (2023): 1-11.

---

> ### Author Response · Authors · 2024-08-10
>
> Dear Reviewer e7ax,
>
> We kindly request your feedback on whether our response has satisfactorily addressed your concerns. If any issues remain or further clarification is needed, please let us know, and we will try to address them before the discussion period ends.
>
> We are looking forward to hearing from the reviewer.
>
> Best regards,
>
> The Authors

---

> > ### Comment · Reviewer_e7ax · 2024-08-13
> >
> > Dear Authors,
> >
> > Thank you for your clarification and additional experiments. Your responses have been helpful, and I have accordingly increased my score.
> >
> > Best

---

> ### Author Response · Authors · 2024-08-14
>
> Dear Reviewer e7ax,
>
> We're pleased to hear that we've addressed your concerns. Thank you for your time and effort!
>
> Best regards,
>
> The Authors

---

### Official Review · Reviewer_Kz4o · 2024-07-15

**Soundness:** 3
**Presentation:** 4
**Contribution:** 3
**Rating:** 6
**Confidence:** 3

**Summary:**

This paper focuses on counterfactual fairness in ML models (regardless of the model), which means that the prediction of the model should not change if the input individual had belonged to a different sub-population. The main contribution of this paper is to provide a theoretical trade-off between the performance of the model and counterfactual fairness.

**Strengths:**

- The paper is very well written and well presented. It is easy to follow and understand the results.
- The main strength of the paper is that it shows if we have access to the ground truth counterfactuals, using an unfair model (regardless of what model it is), we can achieve strong fairness and accuracy (Theorem 3.3). Later, they characterize the excess risk of the Bayes optimal predictor under the counterfactual fairness constraint (Theorem 3.4), which is again insightful.

**Weaknesses:**

- Strong assumptions: access to the ground truth counterfactuals and access to the Bayes optimal predictor. However, they attempt to relax the first assumption slightly in Section 3.3. As I am not an expert in this field, I cannot verify how strong the assumptions (before or after relaxation) are and how practical they are.

**Questions:**

- If A is not binary and is multi-class, what would the results of Theorems 3.3 and 3.4  look like? Specifically, in Theorem 3.3 where $x_a = G^*(x, a, 1-a)$, how can you rewrite this in a multi-class setting?

- Again, in the multi-class case, what would the assumption of theorem 3.6 look like? Do you assume a bound on $G^*$ and $\hat{G}$ for every pair of A’s possible values?

**Limitations:**

Yes.

---

> ### Author Rebuttal · Authors · 2024-08-06
>
> We thank the reviewer for acknowledging the writing and theoretical insight of our paper. We will address each of your concerns and questions below.
>
> >**Assumption of access to ground truth and Bayes optimal predictor**
>
> Access to ground truth counterfactuals, as discussed in Section 6, is a common challenge in the literature of Counterfactual Fairness (including but not limited to the discussed related works in our paper). However, this limitation does not undermine the contribution of this work.
> 1. Similar to the previous works in the literature [1], this assumption is mainly used for theoretical analysis. Notably, we also provide the optimality and inherent trade-offs. Both are important but missing in the literature.
> 2. We carefully studied how violating this assumption, namely using imperfectly estimated counterfactuals, can affect performance, both theoretically (Section 3.3) and empirically (Figure 3, Figure 5 in Section 5, and Figure 1, Figure 2 in the attached PDF in Response to All). We note that *our method outperforms baselines by a large margin, showing notable robustness against such violations.*
>
> Regarding access to Bayes optimal predictor, it would be relatively practical to acquire via ERM [2][3][4]. A practical concern would be the performance of the predictor on the estimated counterfactuals when there is counterfactual estimation error. This is similar to the situation of out-of-distribution generalization. Motivated by this, we propose the Counterfactual Risk Minimization (CRM).
>
> The experiments in our paper validate the effectiveness of PCF-CRM, the practical algorithm we proposed that acts on *estimated counterfactuals* and *ERM-trained predictor*. During the rebuttal, we further add experiments with additional dataset and baselines as shown in the attached pdf in Response to all. Again, these results reveal the effectiveness of our method.
>
>
> [1] Zhiqun Zuo, Mahdi Khalili: Xueru Zhang. Counterfactually Fair Representation. NeurIPS, 2023.
>
> [2] Shalev-Shwartz, Shai, and Shai Ben-David. Understanding machine learning: From theory to algorithms. Cambridge university press, 2014.
>
> [3] Mohri, Mehryar, Afshin Rostamizadeh, and Ameet Talwalkar. Foundations of machine learning. MIT press, 2018.
>
> [4] Jana, Soham, et al. "Empirical Bayes via ERM and Rademacher complexities: the Poisson model." The Thirty Sixth Annual Conference on Learning Theory. PMLR, 2023.
>
> >**Theorem 3.3 and 3.4 in the multi-class case**
>
> In the case of multi-class A, the counterfactual generation mechanism can be written as $x_{a \to \tilde{a}}=G(x,a,\tilde{a})$.
> Then, Theorem 3.3 can be written as
> $$\phi^*_{\textnormal{CF}}(x,a)
>         \triangleq  p(A=a)\phi^*(x,a) + \sum_{\tilde{{a} }\neq a}p(A=\tilde{a})\phi^*(x_{a \to \tilde{a}}, \tilde{a})
>         \in \arg\min_{\phi:\mathrm{TE}(\phi)=0}  E[\ell(\phi(X,A), Y)]$$
>     where $x_{a \to \tilde{a}} = G^*(x,a,\tilde{a})$.
> The proof is similar to the current proof for binary $A$. The key difference is that the expectation over $A$ is now over all possible values of $A$ rather than just two terms. After modifying the Lemma 3.2 to $\phi(x,a)\overset{\text{a.s.}}{=}\phi(x_{a \to \tilde{a}}, \tilde{a}),\quad\forall (x,a,\tilde{a}) $, we can get the multi-class version of Eqn (2) (below Line 490) as
> $  \arg\min_{\phi_0} p(A=a) E_{Y|X=x,A=a}[\ell(\phi_0, Y)] + \sum_{\tilde{a}\neq a} p(A=\tilde{a})E_{Y|X=x_{a \to \tilde{a}},A=\tilde{a}}[\ell(\phi_0, Y)]$. Similarly, it is a convex loss function and we could get the result shown above. Similar ideas could be extended to the other theoretical results since none of them fundamentally rely on A being binary. We use binary $A$ mainly for convenience and simplicity of the notation. We will makes clearer in the revised paper.
>
>
> >**Assumption of theorem 3.6 in the multi-class case**
>
> Yes, the assumption could be written as
> $$ \max_{x, a, \tilde{a}}\|G^*(x_a,a,\tilde{a}) - \hat{G}(x_a,a,\tilde{a})\|_2 \leq \epsilon.$$
>
> We will makes clearer in the revised paper.

---

> ### Author Response · Authors · 2024-08-10
>
> Dear Reviewer Kz4o,
>
> We kindly request your feedback on whether our response has satisfactorily addressed your concerns. If any issues remain or further clarification is needed, please let us know, and we will try to address them before the discussion period ends.
>
> We are looking forward to hearing from the reviewer.
>
> Best regards,
>
> The Authors

---

> > ### Comment · Reviewer_Kz4o · 2024-08-14
> >
> > Thank you for the detailed rebuttal and for addressing my questions. After considering your responses, I have carefully reviewed my initial evaluation and decided to maintain my original score.

---

> ### Author Response · Authors · 2024-08-14
>
> Dear Reviewer Kz4o,
>
> We're pleased to hear that we've addressed your concerns. Thank you for your time and effort!
>
> Best regards,
>
> The Authors

---

### Official Review · Reviewer_pyBD · 2024-07-18

**Soundness:** 3
**Presentation:** 3
**Contribution:** 2
**Rating:** 6
**Confidence:** 2

**Summary:**

This paper focuses on counterfactual fairness (CF), which is a promising framework for evaluating the fairness of machine learning models, and analyzes the trade-off between accuracy and CF. The authors give some theoretical results from the perspective of the Bayes optimal classifier under the perfect CF constraint. They also present theoretical analyses for cases where we can not access the ground truth of the causal structure and Bayes optimal classifier. The theoretical results in this paper were validated by experiments on synthetic and semi-synthetic datasets.

**Strengths:**

- S1. This paper is easy to follow and well-organized. Each experiment seems to clearly correspond to what the authors want to validate.
- S2. In Theorems 3.3 and 3.4, the authors analyze the predictive performance that can be achieved under the constraint of perfect CF from the perspectives of the Bayes optimal classifiers and their excess risk. I think the result that the excess risk of CF is completely characterized by the inherent dependency between Y and A as with the existing group fairness is interesting.
- S3. The authors also give analyses for cases where we can not access the ground truth of the counterfactual generator (i.e., causal structure) and Bayes optimal classifier. Furthermore, they propose a simple learning framework, named CRM, that can be applied to both training from scratch and fine-tuning.

**Weaknesses:**

- W1. I am concerned about the practicality of the proposed framework because the experiments in this paper seem insufficient. All the experiments in this paper seem to consider only one observed feature $X$. While these experiments may be sufficient to just validate the theoretical results, they seem too simple to demonstrate the reality and significance of the proposed framework.
- W2. I am also concerned about the causal graph assumed in this paper. While there may be nothing wrong with assuming a causal graph like Figure 1 to show theoretical statements, I believe the behavior of the proposed framework in the case where this assumption is violated should also be verified by experiments.

**---------- post-rebuttal ----------**

I would like to thank the authors for their comments and clarifications.
The author's rebuttal adequately addressed my concerns.

**Questions:**

- Q1. Can the proposed framework be evaluated by the same experimental setting with [Zuo+ 23]? Their setting includes more complex datasets and causal models than those of this paper, and it seems to be applied to the experimental evaluation for the proposed framework of this paper.
- Q2 (minor, optional). While the authors state "our analysis and our method can be naturally extended to multi-class $A$," I could not find how to do that in the paper including appendix. Is it trivial? And does it have no impact on the theoretical results?
- Q3 (minor, optional). Can the proposed framework be adapted to the path-dependent CF constraint?

[Zuo+ 23] Zhiqun Zuo, Mahdi Khalili: Xueru Zhang. Counterfactually Fair Representation. NeurIPS, 2023.

**Limitations:**

Yes, the authors discussed the limitations and potential impacts of their work in Section 6 and Question 10 of Checklist, respectively.

---

> ### Author Rebuttal · Authors · 2024-08-06
>
> We thank the reviewer for acknowledging the writing quality, the theoretical contributions of our paper, and the importance of our practical method. We will address each of your concerns and questions below.
>
> >**W1,Q1: Experiment Settings**
>
> Thanks for your thought. We would like to first clarify that in our semi-synthetic experiment (Section 5.2), the Law School dataset contains *more than one features*, here notation $X$ is a vector. In addition, the experiment setup here is indeed based on [Zuo+23]. The dataset is generated by a pre-trained VAE that follows more complex causal models than that in our synthetic experiment. We train another VAE to estimate unobserved $U$ and counterfactuals. By doing so we validate the effectiveness of our algorithm when one doesn't have access to the ground truth counterfactuals. We will make these setup clearer in the final manuscript.
>
> To further evaluate the performance of our method, we follow the reviewer’s suggestion and add the experiment on UCI Adult. In this experiment we use DCEVAE from [Zuo+23] to make the causal model more complicated --- the same one as that in Figure 4 of [Zuo+23]. *The proposed method again outperforms other baselines on this task*, as shown in Figure 1 in the attached PDF in Response to All. We will add a more comprehensive study in the final manuscript.
>
> >**W2: Assumptions on causal graph**
>
> The validity of our theoretical analysis holds for all causal models that satisfy the condition given by Assumption 3.1. It is not restricted to the specific graph given in Figure 1, as explained in the footnote on Page 4.  In the paper we present Figure 1 as an illustrative example of how fairness issues can arise from a causal perspective that has been widely used in counterfactual fairness literature [Kusner+ 17][Grari+ 23]. We will modify the caption of Figure 1 to make it more clear in the final manuscript. To test the generality of our method, we experiment with semi-synthetic Law School and Adult (please see above) datasets adapted from [Zuo+23]. Here the causal graphs are designed to be more complicated (Figure 3 and 4 in [Zuo+23]). Experiment results validate the effectiveness of our methods in these situations.
>
> >**Q2: Generalizations to Multi-class A**
>
> In the case of multi-class A, the counterfactual generation mechanism can be written as $x_{a \to \tilde{a}}=G(x,a,\tilde{a})$.
> Then, Theorem 3.3 can be written as
> $$\phi^*_{\textnormal{CF}}(x,a)
>         \triangleq  p(A=a)\phi^*(x,a) + \sum_{\tilde{{a} }\neq a}p(A=\tilde{a})\phi^*(x_{a \to \tilde{a}}, \tilde{a})
>         \in \arg\min_{\phi:\mathrm{TE}(\phi)=0}  E[\ell(\phi(X,A), Y)]$$
>     where $x_{a \to \tilde{a}} = G^*(x,a,\tilde{a})$.
> The proof is similar to the current proof for binary $A$. The key difference is that the expectation over $A$ is now over all possible values of $A$ rather than just two terms. After modifying the Lemma 3.2 to $\phi(x,a)\overset{\text{a.s.}}{=}\phi(x_{a \to \tilde{a}}, \tilde{a}),\quad\forall (x,a,\tilde{a}) $, we can get the multi-class version of Eqn (2) (below Line 490) as
> $  \arg\min_{\phi_0} p(A=a) E_{Y|X=x,A=a}[\ell(\phi_0, Y)] + \sum_{\tilde{a}\neq a} p(A=\tilde{a})E_{Y|X=x_{a \to \tilde{a}},A=\tilde{a}}[\ell(\phi_0, Y)]$. Similarly, it is a convex loss function and we could get the result shown above. Similar ideas could be extended to the other theoretical results since none of them fundamentally rely on A being binary. We use binary $A$ mainly for convenience and simplicity of the notation.
>
> >**Q3: Generalizations to Path-dependent CF**
>
> Under the path-dependent CF constraint, a similar principle will hold: counterfactual pairs need to produce the same outcome. In this case, the counterfactual estimation function will need to be modified to account for the additional intervention. However, we conjecture that similar theoretical results will hold since the core ideas are analogous.
>
>
> [Zuo+ 23] Zhiqun Zuo, Mahdi Khalili: Xueru Zhang. Counterfactually Fair Representation. NeurIPS, 2023.
>
> [Kusner+ 17] Kusner, Matt J., et al. "Counterfactual fairness." Advances in neural information processing systems 30 (2017).
>
> [Grari+ 23] Grari, Vincent, Sylvain Lamprier, and Marcin Detyniecki. "Adversarial learning for counterfactual fairness." Machine Learning 112.3 (2023): 741-763.

---

> ### Author Response · Authors · 2024-08-10
>
> Dear Reviewer pyBD,
>
> We kindly request your feedback on whether our response has satisfactorily addressed your concerns. If any issues remain or further clarification is needed, please let us know, and we will try to address them before the discussion period ends.
>
> We are looking forward to hearing from the reviewer.
>
> Best regards,
>
> The Authors

---

> > ### Comment · Reviewer_pyBD · 2024-08-11
> >
> > Dear Authors,
> >
> > Thank you for your clear response, which has adequately addressed my concerns (especially, W1) and my questions.
> > Thus, I changed my score accordingly.

---

> ### Author Response · Authors · 2024-08-14
>
> Dear Reviewer pyBD,
>
> We're pleased to hear that we've addressed your concerns. Thank you for your time and effort!
>
> Best regards,
>
> The Authors

---

### Author Rebuttal · Authors · 2024-08-06

We thank all reviewers for their time reviewing our paper and providing helpful feedback. In summary, all reviewers acknowledge:
1. The theoretical contributions, especially Theorem 3.3 and Theorem 3.4, which provide an optimality solution under the constraint of Counterfactual Fairness (CF) and characterize the inherent trade-off between CF and predictive performance.
2. The writing quality and result presentation.

We address each reviewer’s questions and concerns in separate responses below. Attached is a PDF containing the experimental results requested by Reviewer pyBD and Reviewer e7ax.
1. In response to Reviewer pyBD, Figure 1 shows results with more complex datasets and causal models. It shows that our method outperforms baselines in this new setup.
2. In response to Reviewer e7ax, Figure 2 shows results with additional baselines. It shows that our method outperforms these new baselines.

---

### Decision · Program_Chairs · 2024-09-25

**Decision:**

Accept (poster)

**Comment:**

The authors demonstrated that, under the assumtption that the Bayes optimal predictor and the ground truth counterfactual generating mechanism are known, it is possible to construct a counterfactually fair Bayes optimal predictor through post-processing.
They also showed that under appropriate assumptions, the use of estimated counterfactual generating mechanism, instead of the true one, could lead to fair predictions under some theoretically guarantee.

While the reviewers expressed concerns about the strength of the assumptions requiring the Bayes optimal predictor and the ground truth counterfactual generating mechanism, the reviewers were generally positive about the paper given that the paper discusses some ways to relax these assumptions.
One of the proposed relaxation methods, counterfactual risk minimization, received detailed questions from the reviewers, and the authors have stated that they will add more detailed discussions on these points.
Incorporating the content of these discussions into the manuscript will improve the quality of the paper.